# Comparison of Wealth-Related Inequality in Tetanus Vaccination Coverage before and during Pregnancy: A Cross-Sectional Analysis of 72 Low- and Middle-Income Countries

**DOI:** 10.3390/vaccines12040431

**Published:** 2024-04-17

**Authors:** Nicole E. Johns, Cauane Blumenberg, Katherine Kirkby, Adrien Allorant, Francine Dos Santos Costa, M. Carolina Danovaro-Holliday, Carrie Lyons, Nasir Yusuf, Aluísio J. D. Barros, Ahmad Reza Hosseinpoor

**Affiliations:** 1Department of Data and Analytics, World Health Organization, 20 Avenue Appia, 1211 Geneva, Switzerland; johnsn@who.int (N.E.J.);; 2International Center for Equity in Health, Federal University of Pelotas, Rua Mal Deodoro 1160, Pelotas 96020-220, Brazil; 3Causale Consulting, Avenida Adolfo Fetter 4331, Pelotas 96090-840, Brazil; 4Department of Immunization, Vaccines, and Biologicals, World Health Organization, 20 Avenue Appia, 1211 Geneva, Switzerland

**Keywords:** health inequality, maternal and neonatal tetanus, immunization, vaccination, health disparities

## Abstract

Immunization of pregnant women against tetanus is a key strategy for reducing tetanus morbidity and mortality while also achieving the goal of maternal and neonatal tetanus elimination. Despite substantial progress in improving newborn protection from tetanus at birth through maternal immunization, umbilical cord practices and sterilized and safe deliveries, inequitable gaps in protection remain. Notably, an infant’s tetanus protection at birth is comprised of immunization received by the mother during and before the pregnancy (e.g., through childhood vaccination, booster doses, mass vaccination campaigns, or during prior pregnancies). In this work, we examine wealth-related inequalities in maternal tetanus toxoid containing vaccination coverage before pregnancy, during pregnancy, and at birth for 72 low- and middle-income countries with a recent Demographic and Health Survey or Multiple Indicator Cluster Survey (between 2013 and 2022). We summarize coverage levels and absolute and relative inequalities at each time point; compare the relative contributions of inequalities before and during pregnancy to inequalities at birth; and examine associations between inequalities and coverage levels. We present the findings for countries individually and on aggregate, by World Bank country income grouping, as well as by maternal and neonatal tetanus elimination status, finding that most of the inequality in tetanus immunization coverage at birth is introduced during pregnancy. Inequalities in coverage during pregnancy are most pronounced in low- and lower-middle-income countries, and even more so in countries which have not achieved maternal and neonatal tetanus elimination. These findings suggest that pregnancy is a key time of opportunity for equity-oriented interventions to improve maternal tetanus immunization coverage.

## 1. Introduction

Tetanus is a potentially life-threatening infection caused by the bacteria *Clostridium tetani*, often present in soil, manure, and agricultural waste. Maternal and neonatal tetanus (MNT) can be a consequence of deliveries and umbilical cord care practices in non-sterile and unsanitary conditions. When tetanus develops, the fatality rates are extremely high, especially when appropriate medical care is not available. Immunization against tetanus for women of reproductive age or during pregnancy is important for protecting both pregnant women and newborns. Maternal and neonatal tetanus elimination (MNTE) has been a goal of the World Health Organization (WHO) and global health partners since the 1980s, aiming to reduce MNT cases to such low levels that the disease is no longer a major public health concern (less than one case of neonatal tetanus per 1000 live births in every district in a country each year) [1]. While progress continues to be made, and MNTE has been achieved by 48 of 59 high-burden countries targeted for the global initiative since the turn of the century, 11 countries had still not reached MNTE status as of December 2023, all of which are low- or lower-middle- income countries [2]. Mali was validated for MNTE in August 2023; report forthcoming.

The burden of MNT is a health equity issue affecting those who experience disadvantage, poverty, and a lack of access to adequate health services [1]. A recent cross-sectional study of household survey data from 76 countries revealed that tetanus immunization protection at birth (PAB) of an infant was highest among mothers who were older, who had higher levels of education, who lived in urban (rather than rural) areas, and who had higher household wealth [3]. While the study found that inequalities had reduced in a ten-year period in six countries amid improvements in overall coverage, it also observed little change in inequalities on aggregate and substantially greater inequalities in coverage among countries which have not achieved MNTE. Reducing inequalities in immunization coverage is a key aim of MNTE programmes, as well as an overall target of global initiatives such as the Immunization Agenda 2030 (IA2030) [4].

Neonatal tetanus is preventable through safe delivery and umbilical cord care practices, as well as through tetanus toxoid-containing vaccines (TTCV), such as tetanus toxoid (TT) or tetanus-diphtheria (Td), which are included in routine immunization programmes globally and administered during antenatal care contacts in many countries. Tetanus protection at birth is a complex coverage metric composed of tetanus immunization during and before the most recent pregnancy (such as receipt of immunization through childhood diphtheria−tetanus−pertussis (DTP) vaccine, booster TTCV doses, mass immunization campaigns of women of reproductive age, or prior pregnancies) [5]. WHO tetanus immunization recommendations have evolved over time, going from targeting pregnant women to targeting women of reproductive age to now recommending a routine six-dose child and adolescent tetanus schedule. Pregnant women and their newborn infants are protected from birth-associated tetanus if the mother received either six doses during childhood or five doses during adolescence/adulthood [5]. However, national immunization schedules differ based on local epidemiology and various programmatic objectives and issues, meaning that they may not always be aligned with the latest WHO recommendations. Decomposing the inequalities in protection at birth to understand whether they are driven by inequalities in coverage before pregnancy, during pregnancy, or in both time periods has important policy and intervention implications. For instance, coverage inequalities that are related to inequalities that appear during pregnancy indicate gaps in antenatal care delivery, access, and utilization [6]. However, to date, there have been no multi-country analyses (nor even small-scale country-level analyses) that have explored this.

Inequalities by time of vaccination have been explored for other vaccination types to support the identification of potential health system gaps. For instance, inequalities in childhood receipt of the DTP vaccine can be broken down by receipt of the first DTP vaccine dose (DTP1) and the third DTP vaccine dose (DTP3) to provide insight into the performance of the vaccine delivery system. Inequalities in DTP1 coverage (recommended at around 6 to 8 weeks after birth) indicate systemic challenges with access to and utilization of child health services and the need for general health system strengthening, while inequalities in DTP3 coverage (recommended at around 14 weeks or 6 months of age, depending on the country) are a signal of health service quality and other barriers experienced by mothers or caregivers [7,8,9].

Data on the receipt of tetanus immunization during pregnancy and coverage at birth are available in Demographic and Health Surveys (DHS) and Multiple Indicator Cluster Surveys (MICS), which are nationally representative household surveys carried out in several low- and middle-income countries. In this study, we quantify the extent of wealth-related inequality in tetanus immunization coverage before pregnancy versus during pregnancy, summarizing the amount of inequality introduced or mitigated during pregnancy and exploring variation by country, country income level groupings, and MNTE status.

## 2. Materials and Methods

### 2.1. Data Sources

Data for this study come from recent DHS and MICS, which collect a wide range of information regarding health and other topics in low- and middle-income countries (extensive information on their methodologies has been published elsewhere) [10,11]. All publicly available surveys conducted within the prior 10 years at time of analysis (2013–2022) were considered for inclusion. Surveys were excluded if they did not contain the outcome measure of interest (defined below, resulting in exclusion of *n* = 21 countries), or if estimates stratified by the dimension of inequality could not be produced due to small numbers of respondents within one or more levels of the inequality dimension (defined as <25 individuals, resulting in exclusion of *n* = 4 countries). When multiple surveys from the same country were available within the study time range, the most recent survey was selected, as the aim of this research is to characterize the most recent state of inequality in maternal tetanus immunization coverage.

All data were processed by the International Center for Equity in Health (ICEH, www.equidade.org) at the Universidade Federal de Pelotas. All outcome measures used in analyses, detailed below, were calculated by the ICEH directly from raw survey data.

### 2.2. Outcome Measures

Maternal tetanus immunization coverage at the time of birth of the infant can be received either before or during pregnancy (or in the case of multiparous women, the most recent pregnancy). It, in turn, provides tetanus protection to the infant (i.e., coverage at birth):*Coverage before pregnancy + Coverage during pregnancy = Coverage at birth*

Information about tetanus immunization coverage during pregnancy and coverage at birth was available directly in the survey data. Coverage before pregnancy was calculated as the arithmetic difference of these two measures:*Coverage at birth − Coverage during pregnancy = Coverage before pregnancy*

*Coverage of tetanus at birth*: This indicator is defined as the proportion of women aged 15–49 years who had a live birth within the five years (for DHS) or two years (for MICS) preceding the survey and who received one of the following: (a) Two tetanus toxoid-containing vaccine (TTCV, tetanus toxoid—TT or tetanus-diphtheria—Td) doses during the pregnancy for her most recent live birth; (b) two or more TTCV doses before the last pregnancy (the last within 3 years of the most recent live birth); (c) three or more TTCV doses before the last pregnancy (the last within 5 years of the most recent live birth); (d) four or more TTCV doses before the last pregnancy (the last within 10 years of the most recent live birth); or (e) five or more TTCV doses at any time prior to the most recent live birth.

*Coverage of tetanus during pregnancy:* This indicator is defined as the proportion of women aged 15–49 years who had a live birth within the five years (for DHS) or two years (for MICS) preceding the survey and who received two or more TTCV doses during pregnancy.

Data collection for these indicators is restricted to the lastborn child in both DHS and MICS surveys. Data is obtained by maternal recall for both surveys, though both ask to see vaccination cards where available.

This analysis was limited to women who had only one live birth (i.e., for whom the birth reported in the survey was their first live birth). In women who have had multiple pregnancies, coverage before the current pregnancy could have occurred (a) in childhood, (b) through other adolescent/adult immunization outside of pregnancy, or (c) during prior pregnancies. We chose to limit the analyses to women for whom the survey data related to a first birth to aid in interpretation of the coverage before the pregnancy time period, as this population restriction largely eliminates the possibility of vaccination occurring during previous pregnancies.

### 2.3. Dimension of Inequality

We assess wealth-related inequality in tetanus immunization coverage using country-specific household wealth quintiles. Household wealth is derived from household asset indices and is directly provided in DHS and MICS datasets [12]. The households, weighted by size, are divided into five equal groups, or wealth quintiles, each representing 20% of the population.

### 2.4. Statistical Analyses

We start by summarizing tetanus immunization coverage at the three time points of interest (before pregnancy, during pregnancy, and at birth) for the study sample overall. We then present the percentage of coverage occurring during pregnancy, overall and by country World Bank income grouping at time of survey (low income, lower-middle income, and upper-middle income) [13]. Population-weighted mean values and corresponding 95% confidence intervals are presented for aggregate group estimates.

We measure absolute wealth-related inequalities in tetanus immunization coverage at each time point using the slope index of inequality (SII) and relative inequality using the relative index of inequality (RII) [14]. The SII is calculated via population-weighted logistic regression of coverage across five wealth quintiles ranked from the least to most wealthy, ultimately indicating the absolute difference in the predicted coverage between the wealthiest and least wealthy subgroups. We multiplied the SII by 100 to facilitate interpretation in percentage points, and thus the SII values presented here range from −100 to 100. A positive SII value indicates that coverage is higher among the wealthiest, while a negative SII indicates the opposite; a value of 0 for SII represents equity. The RII is calculated similarly and represents the ratio of the coverage predicted for the wealthiest quintile divided by the coverage predicted for the least wealthy quintile. A RII value greater than 1 suggests that coverage is higher among the wealthiest, while a RII value less than 1 suggests that coverage is higher among the least wealthy and a value of 1 for RII represents equity.

For both absolute and relative inequality, and for both before pregnancy and during pregnancy time periods, we summarize countries with inequality favoring the wealthiest quintile (statistically significant SII > 0, RII > 1), relatively equitable coverage (SII with 95% confidence interval including 0; RII with 95% confidence interval including 1), or inequality favoring the least wealthy quintile (statistically significant SII < 0, RII < 1). We then test the association between inequality before versus during pregnancy via Pearson correlation coefficients.

Next, we summarize the SII during pregnancy to show the amount of absolute inequality introduced or mitigated during pregnancy. We present SII by country, as well as weighted mean estimates for the sample overall and by country income grouping. We characterized countries as having a meaningful increase in absolute inequality favoring the wealthiest (statistically significant SII during pregnancy > 0), similar inequality (SII during pregnancy statically equivalent to 0), or a meaningful decrease in inequality favoring the wealthiest (statistically significant SII during pregnancy < 0). However, because inequality is bimodal in SII (−100 = total inequality favoring the least wealthy on one end, 100 = total inequality favoring the wealthiest on the other), a negative value for a SII during pregnancy could actually reflect an increase in inequality favoring the least wealthy. Thus, we stratify by the nature of inequality before pregnancy to additionally categorize countries with inequality favoring the least wealthy before pregnancy and a negative SII during pregnancy as ‘a meaningful increase in inequality favoring the least wealthy’.

We then assess whether there is an association between SII during pregnancy with the level of tetanus immunization coverage at birth, summarizing the patterns of coverage and inequality.

Finally, we summarize inequality for countries which have and have not achieved maternal and neonatal tetanus elimination at the time of survey.

As a post hoc sensitivity analysis, we examine whether key findings are sensitive to the recency of data availability, summarizing coverage and inequality separately for countries with data available within the past five years (2019–2022) and countries with data greater than five years old (2013–2018).

Statistical analyses were conducted using Stata Statistical Software version 18.0, College Station, TX [15]. All country-level point estimates and uncertainty estimates took into account relevant survey sampling design and survey weights using the *svy* command; for calculation of standard errors, strata with a single primary sampling unit were centered at the grand mean via *singleunit (centered)* specifications. Multi-country average estimates were weighted based on the national population of women of reproductive age (15–49) from UN World Population Prospects 2022 data [16]. Significance was set at *p* = 0.05 for all comparisons; 95% confidence intervals (CIs) are reported throughout.

Ethical clearance for surveys analyzed in this study was obtained through the responsible institutions that administered the surveys. Each of the 72 included surveys underwent individual review from a relevant in-country ethical review board, and additional detail can be found in individual survey final reports. In addition, all DHS surveys were reviewed and approved by the institutional review board of ICF, the organization which oversees DHS implementation [17]. All analyses presented here were conducted using anonymized databases, ensuring the protection of privacy and confidentiality.

## 3. Results

### 3.1. Sample Characteristics

The final analytic sample included 72 low- and middle-income countries with a recent DHS or MICS (2013–2022) containing measures of tetanus immunization coverage and wealth quintiles. The total sample included 158,753 women who had their first live birth in the five (for DHS) or two (for MICS) years prior to the survey. Our sample included 21 low-income countries, 33 lower-middle-income countries, and 18 upper-middle-income countries at time of survey. Additionally, all tetanus protection occurred during pregnancy in one country (Chad). As this means that coverage before pregnancy was 0% for all wealth quintiles, the SII and RII before pregnancy could not be calculated; thus, the sample for all analyses involving measures of inequality before pregnancy is 71 countries.

Tetanus immunization schedules vary widely across settings. The WHO recommends a six-dose schedule, including three doses in infancy and one dose each at 12–23 months, 4–7 years, and 9–15 years. At the time of survey, all countries’ immunization schedules included a three-dose TTCV series in infancy (see Appendix A). Only a third of countries (26 of 72) included a booster dose at age 12–23 months, 29 countries included a booster dose at age 4–7 years, and 19 countries included a booster dose at age 9–15 years. About half of countries (37 of 72) did not include any boosters between the infant series and pregnancy. Only a fifth of countries (15 of 72) included the full recommended six-dose TTCV series in their immunization schedule at time of survey. Note, however, that women of reproductive age at time of survey would have received these recommended booster doses based on prior schedules in place during their childhood and adolescence, or through mass vaccination campaigns happening outside of the standard schedule.

### 3.2. Tetanus Immunization Coverage before Pregnancy, during Pregnancy, and at Birth

Tetanus coverage at each time point varied widely across countries (See Table 1, Supplementary Appendix A). Coverage before pregnancy was 10% [95% CI 7–12%] on average in our study population, ranging from 0% in Chad to 51% in Bangladesh. Coverage during pregnancy was higher at 67% [95% CI 62–72%] on average, ranging from 9% in Suriname to 88% in India. Coverage at birth, the sum of these two coverages, averaged 76% [95% CI 72–80%] and ranged from 15% in Suriname to 93% in India (See Figure 1 and Figure 2).

Immunization coverage at birth increased monotonically with increasing wealth overall and for low- and lower-middle-income countries. This is due largely to increases in coverage during pregnancy with increasing wealth (See Figure 3).

Overall, tetanus coverage at birth is comprised mostly of coverage during the recent pregnancy. In 66 of 72 countries, the majority of coverage (more than 50%) occurred in pregnancy. Across countries, the average percentage of PAB coverage occurring in pregnancy was 86% [95% CI 83–90%], ranging from 20% of total PAB coverage occurring in pregnancy in Costa Rica to 100% of total PAB coverage occurring in pregnancy in Chad. The percentage of coverage occurring during pregnancy was significantly higher among low- and lower-middle-income relative to upper-middle-income countries: low-income countries had an average 86% [95% CI 83–89%] of coverage occur during pregnancy; lower-middle-income countries had 88% [95% CI 82–93%] of coverage during pregnancy; while upper-middle-income countries had 76% [95% CI 71–81%] of coverage during pregnancy.

### 3.3. Inequality in Tetanus Immunization Coverage before Pregnancy, during Pregnancy, and at Birth

On the whole, there was greater absolute inequality in tetanus immunization coverage (measured via SII) favoring the wealthiest quintile during pregnancy compared to before pregnancy (See Table 2). The SII before pregnancy averaged 2.4 percentage points [95% CI 1.0–3.7], ranging from −17.5 percentage points in Suriname to 35.1 percentage points in Zambia; the SII during pregnancy averaged 10.5 percentage points [95% CI 5.3–15.6], ranging from −24.9 percentage points in Iraq to 64.3 percentage points in Nigeria; and the SII at birth averaged 12.8 percentage points [95% CI 7.7–17.9], ranging from −24.4 percentage points in Suriname to 66.3 percentage points in Nigeria.

Relative inequality (measured via RII) was equivalent with regard to the direction of inequalities at each time point. In contrast to absolute inequality, however, we observed similar magnitudes of relative inequality before versus during pregnancy (See Table 3). The RII before pregnancy averaged 1.4 [95% CI 1.1–1.6], ranging from 0.04 in Suriname to 6.4 in Mali; the RII during pregnancy averaged 1.4 [95% CI 1.1–1.6], ranging from 0.14 in Costa Rica to 10.2 in Yemen; and the RII at birth averaged 1.3 [95% CI 1.2–1.5], ranging from 0.18 in Suriname to 3.7 in Papua New Guinea.

The distribution of countries by presence and direction of absolute inequalities at each time point is summarized in Figure 4.

### 3.4. Inequalities in Coverage before Versus during Pregnancy

There is no clear single pattern of the relationship between absolute inequalities in coverage before versus during pregnancy (See Figure 5). There was not a significant correlation between SII before vs. SII during pregnancy, for the sample overall, nor by country income group (overall r = −0.18, *p* = 0.14). A third of countries (24 out of 71) had little to no inequality in coverage before pregnancy and inequality favoring the wealthiest quintile during pregnancy. A quarter of countries (19 out of 71) had little to no inequality at both time points. Ten percent (7 of 71) had inequality favoring the wealthiest at both time points. Patterns of inequality before and during pregnancy are similar when examining relative rather than absolute inequality (See Appendix A).

### 3.5. Change in Absolute Inequality from before Pregnancy to Birth

The value of the SII during pregnancy reflects the change in absolute inequality that was introduced or mitigated during pregnancy (See Figure 6). Half of countries (35) had a meaningful increase in inequality favoring the wealthiest in tetanus immunization coverage during pregnancy; 28 countries had similar inequality, eight countries had a meaningful decrease in inequality favoring the wealthiest, and only one country had a meaningful increase in inequality favoring the least wealthy.

Overall, we observe inequality favoring the wealthiest during the pregnancy time period, with the largest increases in inequality in low-income countries. Upper-middle-income countries, in contrast, had no collective change in inequality during the pregnancy time period. The average SII in pregnancy for the overall sample was 10.5 percentage points [95% CI 5.3–15.6], 25.4 [95% CI 19.8–31.0] for low-income countries, 10.0 [95% CI 2.1–17.8] for lower-middle-income countries, and −3.8 [95% CI −8.7–1.0] for upper-middle-income countries.

Observed increases in absolute inequality favoring the wealthiest during pregnancy were coupled with increases in tetanus immunization coverage during this period, both overall and in low- and lower-middle-income countries (See Figure 7 and Figure 8). However, absolute inequality in pregnancy was not significantly associated with coverage in pregnancy for the sample overall, nor by country income groupings (overall r = 0.21, *p* = 0.07). Furthermore, several countries had meaningful decreases in absolute inequality and increases in coverage during pregnancy, despite the overall trend. Eight countries—Bangladesh, Costa Rica, Cuba, Egypt, Iraq, Namibia, Nepal, and the Philippines—all had statistically significant decreases in absolute inequality favoring the wealthiest during pregnancy of five percentage points or more, coupled with increases in immunization coverage.

### 3.6. Absolute Inequality and Maternal and Neonatal Tetanus Elimination Status

Coverage and equity in tetanus immunization is of particular relevance for those countries which have not yet achieved maternal and neonatal tetanus elimination (MNTE) (defined as less than one case per 1000 live births in every district of a country); we therefore examine differences in absolute inequality between countries that had achieved or had not achieved MNTE at time of survey. We have data from nine countries which had not achieved MNTE as of 2023 (Afghanistan, Angola, Central African Republic, Guinea, Nigeria, Pakistan, Papua New Guinea, Sudan, and Yemen), as well as data from five countries where MNTE was achieved after the most recent available survey (Chad, Democratic Republic of the Congo, Ethiopia, Haiti, and Mali). Average TTCV coverage was significantly lower among countries which had not achieved MNTE relative to countries which had achieved MNTE before pregnancy (6.0% vs. 10.5%, *p* = 0.03), during pregnancy (58.6% vs. 68.7%, *p* = 0.04), and at birth (64.6% vs. 79.1%, *p* < 0.001).

Absolute inequality in tetanus immunization coverage during pregnancy was significantly higher among countries which had not achieved MNTE compared to those which had achieved elimination (See Figure 9). The average SII before pregnancy was 4.8 percentage points [95% CI 1.3–8.3] in countries that had not achieved MNTE compared to 1.8 percentage points [95% CI 0.3–3.2] in countries that had achieved MNTE (*p* = 0.12). The average SII during pregnancy was 45.0 percentage points [95% CI 36.3–53.7] in countries that had not achieved MNTE compared to 2.1 percentage points [95% CI −1.3—5.5] in countries which had achieved MNTE (*p* < 0.001). Nearly all countries which had not achieved MNTE (13 of 14) had significant inequality of favoring the wealthiest during pregnancy.

### 3.7. Sensitivity Analysis—Recency of Data Availability

To examine whether findings from this study were sensitive to the recency of data availability, we replicate key analyses separately for countries with data available from the past five years (2019–2022, *n* = 28) and countries with data greater than five years old (2013–2018, *n* = 44). The results indicate that the majority of tetanus immunization coverage at birth occurs during pregnancy in most countries with recent data (25/28, 89%) as well as in most countries with older data (41/44, 93%). We find significantly lower average absolute wealth-related inequality in the more recent data before pregnancy (average SII for recent data 0.8 vs. 4.9 for older data, *p* = 0.003) and significantly lower average inequality at birth (average SII for recent data 8.7 vs. 19.3 for older data, *p* = 0.04), though we find no significant difference in average inequality during pregnancy (*p* = 0.20). We find consistent patterns of inequality whereby a similar proportion of countries have statistically significant inequalities before pregnancy, during pregnancy, and at birth (See Figure 10).

In both the more recent and older data, we observe significant average inequality favoring the wealthiest during the pregnancy time period (average SII for recent data 7.9, 95% CI 0.3–15.5; average SII for older data 14.6, 95% CI 7.4–21.8). Note that these findings should not imply temporal trends, as each time period contains a unique group of countries. However, consistent findings of significant inequality during pregnancy and consistent evidence of greater inequality during pregnancy compared to before pregnancy suggest that the overall conclusions of the study are not sensitive to a 5-year rather than 10-year analysis time frame.

## 4. Discussion

Findings from this study of 72 low- and middle-income countries suggest that the majority of tetanus immunization PAB for first births is the result of TTCV doses received during pregnancy. This study also highlights absolute wealth-related inequality in tetanus immunization PAB, finding that the majority of this inequality is introduced during pregnancy rather than before it. This is particularly evident in low- and lower-middle-income countries.

In 92% of countries examined in this study (66 of 72), PAB appears to be driven by immunizations received during pregnancy, with limited coverage pre-pregnancy. The World Health Organization formally recommends coverage of six doses of TTCV for all people, including the 3-dose primary series received in childhood and three boosters given at 12–23 months, 4–7 years, and 9–15 years of age [5]. Adherence to this schedule would result in all women of reproductive age having PAB prior to any pregnancies. However, only 21% of examined countries had adopted this schedule at the time of survey, and it’s likely that even fewer had this schedule in place at the time the women surveyed were children/adolescents and eligible for these booster doses. This ultimately results in a reliance on the pregnancy time period to provide sufficient TTCV doses to protect the neonate and mother by the time of birth. As more countries shift to the six-dose schedule, the relative contributions of coverage before and during pregnancy will likely shift. However, the data examined here suggest that coverage in pregnancy remains the primary input to PAB as of these recent surveys, and thus warrants ongoing attention.

Though tetanus immunization before pregnancy was low overall, and TTCV doses received in pregnancy increased PAB coverage, this increase in coverage was accompanied by increased inequality in coverage in half of the countries we examined. Thirty-five out of 71 countries had a meaningful increase in inequality favoring the wealthiest in tetanus immunization coverage from pre-pregnancy to birth. Absolute inequality was markedly larger during pregnancy compared to pre-pregnancy for the overall sample (SII 10 versus 2 percentage points). Though countries have the same statistically significant inequalities at each time point when using a relative rather than absolute measure of inequality, the relative inequality measure suggests a similar magnitude of inequality before versus during pregnancy. This indicates that the large differences in absolute inequality during versus before pregnancy are driven in part by the large amount of coverage occurring in pregnancy.

Inequalities in TTCV receipt during pregnancy are likely driven in part by inequalities in antenatal care (ANC) access and utilization [18,19]. Other studies have observed increases in uptake of ANC services alongside increased inequalities in service utilization favoring the wealthiest in low-income settings [20]. However, higher coverage of ANC visits relative to coverage of TTCVs, particularly in settings which have not achieved MNTE, suggests ongoing missed opportunities for vaccination within routine pregnancy care [21,22,23,24]. Other studies have shown inconsistent trends in the utilization of tetanus vaccination among pregnant women in the context of antenatal care over time [25]. Additionally, inequalities in healthcare utilization may result in inequalities in tetanus protection beyond that conveyed by vaccination alone; specifically, facility delivery and skilled birth attendance are associated with greater use of safe and sterile delivery and umbilical cord care practices, as well as subsequent reductions in tetanus mortality [26]. As such, inequalities in healthcare utilization during pregnancy and birth may contribute to inequalities in tetanus protection beyond inequalities in immunization alone. Importantly, inequalities in ANC uptake among pregnant women also signal adverse outcomes for early childhood among their children. For example, lower levels of ANC visits among pregnant women have also been shown to be associated with incomplete vaccination among their children in early childhood [27]. Findings from our study support equity-oriented interventions and policies targeted towards pregnant women both within and outside of routine healthcare, such as additional education for healthcare providers on the importance of screening for, providing, and recording TTCV doses during ANC visits, or supplemental immunization activities (SIAs) focused on geographic areas at greatest disadvantage [21,28].

In low-income countries in particular, inequalities in coverage at birth were introduced during pregnancy. While only 4 of 21 examined low-income countries had significant inequality favoring the wealthiest in tetanus immunization coverage before pregnancy, 16 of 21 had significant inequality during pregnancy. On aggregate, the average absolute inequality among low-income countries was notably higher during pregnancy (SII 25 percentage points) compared to before pregnancy (SII six percentage points). Conversely, the majority of upper-middle-income countries in our study had negligible inequality during pregnancy (11 of 18), and, on average, had a decrease in inequality during pregnancy (SII-4 percentage points). Upper-middle-income countries also had notably different patterns of tetanus PAB coverage relative to low- and lower-middle-income countries. A significantly greater proportion of PAB coverage occurred before pregnancy, or, in other words, significantly less PAB coverage occurred during pregnancy. Additionally, PAB coverage was highest among middle-wealth mothers (3rd quintile), with lower coverage among both the least and most wealthy quintiles, while in low- and lower-middle-income countries, PAB coverage increased monotonically with increasing wealth.

The divergent patterns of coverage and inequality observed in this study among upper-middle-income countries are likely driven by several factors. First, the greater coverage before pregnancy highlights better access to and implementation of immunization services throughout the life course prior to pregnancy [29], likely reflecting better documentation of immunizations, as discussed in further detail below. The pattern of highest PAB coverage among middle-wealth women, driven by higher levels of coverage in pregnancy, might be explained by an indirect relationship between tetanus vaccination and better birth and antenatal care conditions. As the coverage of institutional delivery increases and the structure of services also increase, less attention might be applied to tetanus immunization, as safe birth and umbilical cord care conditions are present. Meanwhile, the wealthiest women are likely accessing the highest quality of birth care, where prevention of neonatal tetanus may no longer be a priority concern. Such findings align with previous work examining institutional deliveries [30]. Analyses which jointly consider tetanus immunization and measures of care utilization such as institutional delivery are beyond the current study’s scope but would help in further understanding this potential relationship.

In addition to differences across country income groupings, we see striking differences in inequalities between countries that had and had not achieved MNTE. Countries which had not achieved MNTE at time of survey had a 20 times higher average absolute inequality in coverage compared to countries which had achieved MNTE (SII 45 vs. 2 percentage points). The large amount of absolute inequality in coverage within countries that had not achieved MNTE is expected, as the ongoing presence of neonates who develop NT requires a substantial population of mothers who lack immunization coverage and who face exposure to tetanus through unsafe birth or umbilical cord care practices. MNTE efforts that seek to identify these populations routinely find that areas with greatest risk are rural, remote, and economically unstable [31]. MNTE strategies typically include supplemental immunization activities (SIAs) aiming to serve these groups, which may quickly improve coverage and reduce inequality in coverage before pregnancy but will not reduce during-pregnancy inequalities [32]. All countries which had not achieved MNTE were classified as low or lower-middle income countries at time of survey; some of the trends in coverage and inequality observed may have thus been driven by trends in country income level and related factors, such as existing health system structure, rather than MNTE achievement status specifically. However, if we limit the comparison to low- and lower-middle income countries which have achieved MNTE, we find the same significant results, in that the average SII during pregnancy was 45 percentage points in countries that had not achieved MNTE compared to three percentage points in low- and lower-middle-income countries which had achieved MNTE, *p* < 0.001 (results not shown).

Despite overall trends, several countries reduced inequality or kept it low during pregnancy, which can yield important lessons. Eight countries—Bangladesh, Costa Rica, Cuba, Egypt, Iraq, Namibia, Nepal, and the Philippines—all had statistically significant decreases in absolute inequality favoring the wealthiest during pregnancy of five percentage points or more. In Nepal, for example, the Safe Delivery Incentive Programme was implemented in 2005 to include reimbursement for delivery travel costs for all Nepali women and free delivery care and healthcare facility delivery cost reimbursement for women in lower-income districts; this program was shown to significantly improve prenatal care visit and tetanus vaccination uptake while reducing wealth-based inequalities in coverage at the district level [33,34]. In Bangladesh, the use of antenatal services has been steadily increasing [35], and women across socioeconomic groups have shown similar patterns in the utilization of public and private health facilities throughout pregnancy [36]. Better understanding of the country-specific contexts, including tetanus immunization schedules, standards of care for pregnancy and childbirth, as well as other related policies and interventions, could be leveraged to inform efforts to improve tetanus immunization coverage and simultaneously reduce inequality during pregnancy.

Findings from this study should be viewed in light of several limitations. First, though we have limited analyses to first births, a small number of women may have received tetanus vaccines during a prior pregnancy which ended in a miscarriage or abortion, so some caution should be exercised with interpretation of the before-pregnancy coverage time period. However, most miscarriages and abortions occur during the first trimester (Week 12 or earlier) [37,38], before women in pregnancy would typically receive tetanus toxoid (national schedules vary, but generally in weeks 13+), so the number of women this applies to is likely small. Additionally, limiting the sample to first births likely underestimates PAB for births as a whole, as each pregnancy event provides an opportunity for interaction with the health system and an opportunity to receive any necessary or missing TTCV doses. Indeed, an analysis using a similar set of country years of data found a slightly higher median PAB (69.1% vs. 67.5% observed in this study); PAB estimates should thus be considered reflective of first births only [3].

Second, PAB from TTCV doses received prior to pregnancy may be derived from a number of immunization sources. TTCVs received prior to pregnancy include standard childhood vaccine doses, adolescent and adult boosters, women of reproductive age-specific boosters, and doses received as part of SIAs. MNTE initiatives in particular have historically included SIA campaigns targeted at women of reproductive age. For the analyses presented here, we grouped all these sources of pre-pregnancy immunization. As childhood tetanus vaccination coverage differs widely across the examined settings, the relative contributions of these pre-pregnancy immunization sources will also differ substantially by country. Additionally, average age at first birth also varies substantially across countries, affecting how long women have the opportunity to receive needed adult boosters prior to pregnancy and the length of time since early childhood and adolescent doses most subject to recall bias. Any country-specific application of findings should take into consideration the broader fertility and immunization system context in that country, inclusive of childhood immunization coverage and evolving tetanus vaccination strategies and schedules, to generate the most appropriate conclusions for that setting. The limitations in recording and maintaining records of immunization during the life course may also have resulted in misclassification of vaccination status and the contribution of each immunization delivery strategy [39,40]. Future work that disentangles the relative contribution to PAB of each of these sources would be necessary to inform more specific targets to improve PAB coverage and equity prior to pregnancy.

Third, where immunization monitoring and tracking systems are not reliably available at the individual level, women may receive additional and ultimately unnecessary doses of TTCV in pregnancy as a result of poor documentation and recall rather than true need for immunity. The World Health Organization recommends that countries which have not achieved MNTE provide two doses of TTCV to any pregnant women “for whom reliable information on previous tetanus vaccinations is not available” [5]. Conversely, women who do not require additional doses at birth because they were fully immunized before delivery might have this information inaccurately recorded in their home-based or hospital records, or may report it incorrectly in surveys [39]. Consequently, these women could be mistakenly classified as lacking PAB despite the absence of birth doses being justified. Issues of recall also differentially affect TTCV doses received prior to and during pregnancy; as pregnancy doses are more recent, they are by definition less subject to recall bias, as well as more likely to be captured by electronic records and immunization documentation, which are improving over time in most settings. Findings suggesting greater TTCV coverage during pregnancy compared to before pregnancy must be viewed in light of this limitation, though findings regarding inequality within each time frame are less sensitive to this recall bias. Increasing use of electronic records and better immunization documentation and monitoring should lead to a decrease in use of extraneous pregnancy TTCV doses and an increase in accurate reporting of coverage received prior to birth.

Fourth, this research is subject to the limits of the available data, including only a sample of low- and middle-income countries with nationally representative surveys and data which may be up to 10 years old. The coverage and timing of TTCV dose receipts are also subject to immunization card ownership or recall. Patterns of coverage and inequality may have changed since the time of survey, and survey estimates may under-represent true coverage. The sensitivity analyses conducted to examine data collected within five years and greater than five years ago suggest that the conclusions of these analyses are not sensitive to a five-year rather than ten-year analysis time frame; however, findings may still be changing over time. In particular, the COVID-19 pandemic has had large impacts on immunization systems and healthcare service delivery generally, and children, adolescents, and women who missed TTCV doses due to the pandemic or resultant health system impacts should be targeted for catch-up doses to sustain coverage and minimize inequality. Studies which replicate these analyses using the most up-to-date data in a given context will be most valuable for elucidating the current state of inequality.

Finally, the analyses presented here are cross-sectional and limited to the examination of a single dimension of inequality, namely household wealth. Other dimensions, such as maternal age, maternal education, or urban/rural residence, may also be meaningful determinants of tetanus immunization coverage and, when considered jointly, may in fact be greater drivers of inequality than wealth alone. It is beyond the scope of these analyses to examine additional determinants of coverage, dimensions of inequality, or the relative contribution of multiple dimensions of inequality, and future work in these areas would allow for greater understanding of the full complexities of inequalities in immunization coverage.

## 5. Conclusions

In this study of first births among women in 72 low- and middle-income countries, we find that most tetanus PAB coverage is the result of TTCV doses received during pregnancy. We present evidence of significant inequality favoring the wealthiest in PAB coverage, finding that most of this inequality is introduced during pregnancy rather than before it. This is particularly evident in low- and lower-middle-income countries, whereas upper-middle-income countries have greater wealth-related inequality in tetanus immunization coverage during pregnancy. Efforts to ensure high PAB coverage levels at the population level, particularly those taking place during pregnancy, should also consider equity in coverage as a key goal and outcome.

## Figures and Tables

**Figure 1 vaccines-12-00431-f001:**
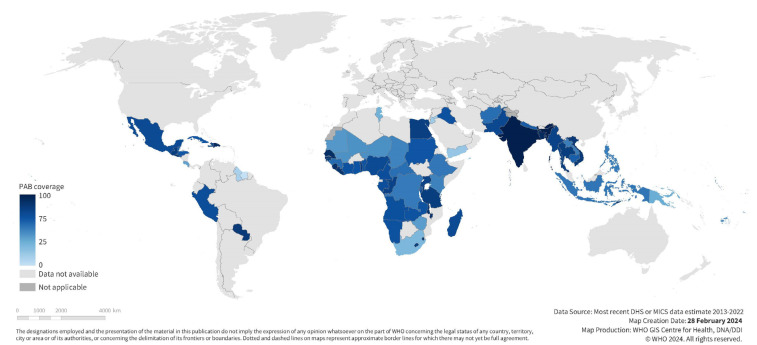
National maternal tetanus immunization coverage at birth among first births; most recent DHS or MICS estimates 2013–2022 for 72 included study countries.

**Figure 2 vaccines-12-00431-f002:**
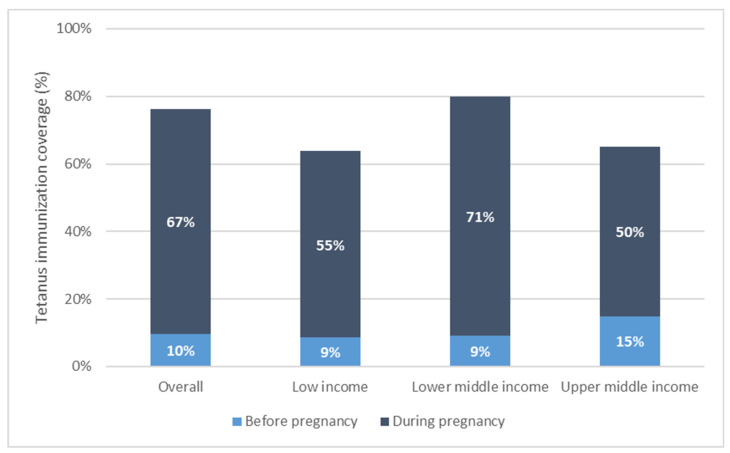
Maternal tetanus immunization coverage by country income grouping, weighted mean coverage level.

**Figure 3 vaccines-12-00431-f003:**
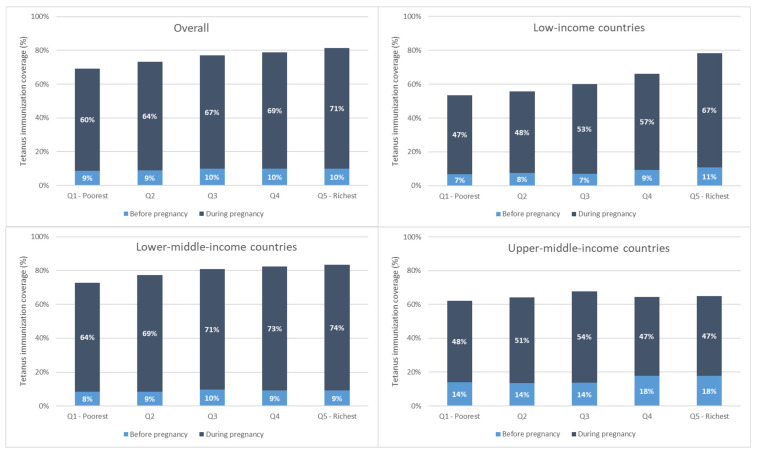
Maternal tetanus immunization coverage by wealth quintile, weighted mean, overall, and by country income grouping.

**Figure 4 vaccines-12-00431-f004:**
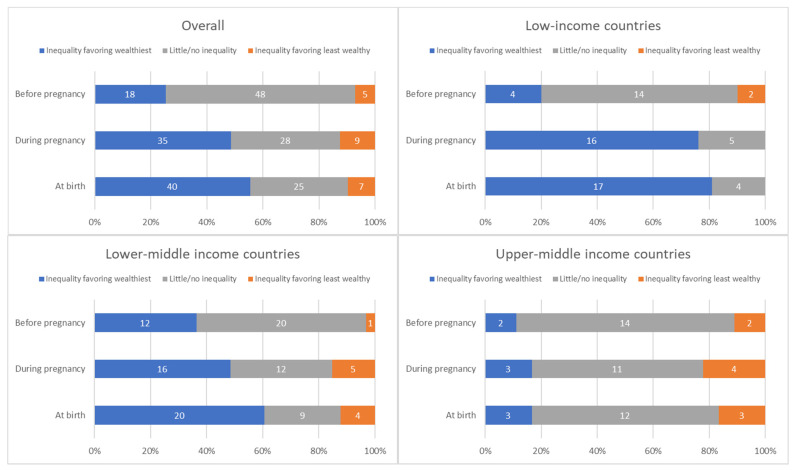
Distribution of countries’ absolute inequality in tetanus immunization coverage before pregnancy, during pregnancy, and at birth; overall and by country income grouping.

**Figure 5 vaccines-12-00431-f005:**
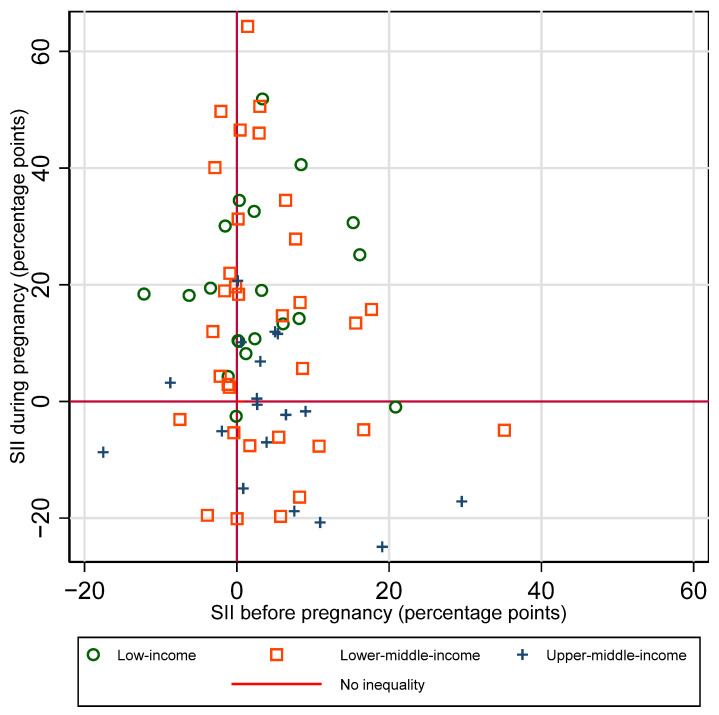
Absolute inequality (SII) in maternal tetanus immunization coverage by wealth quintile, before versus during pregnancy.

**Figure 6 vaccines-12-00431-f006:**
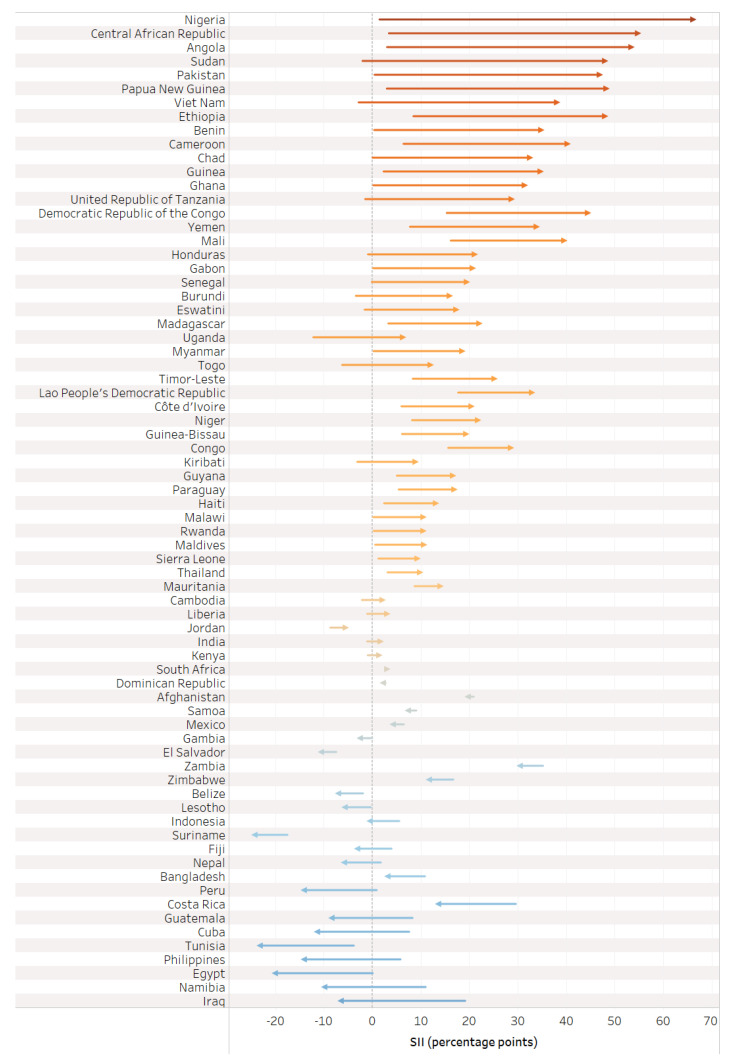
Change in absolute inequality in tetanus immunization coverage before pregnancy versus at birth.

**Figure 7 vaccines-12-00431-f007:**
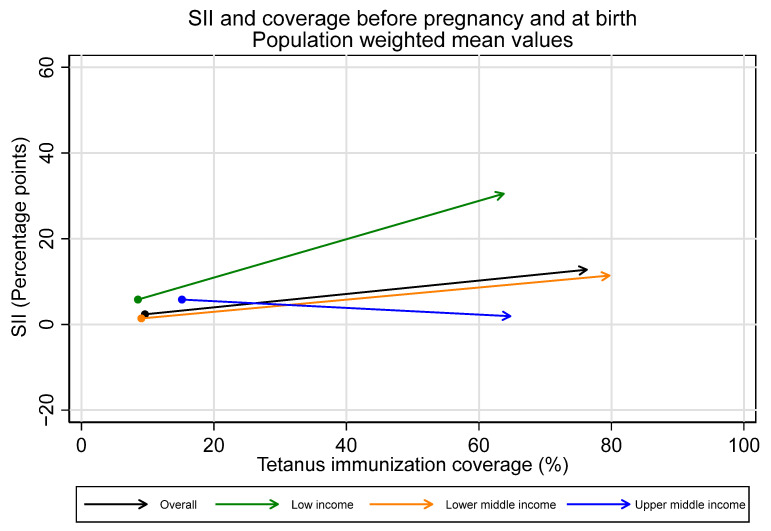
Absolute inequality in maternal tetanus immunization coverage and average coverage level, before pregnancy and at birth. (Dot represents coverage level and SII in tetanus immunization coverage before pregnancy; arrowhead represents coverage and SII at birth).

**Figure 8 vaccines-12-00431-f008:**
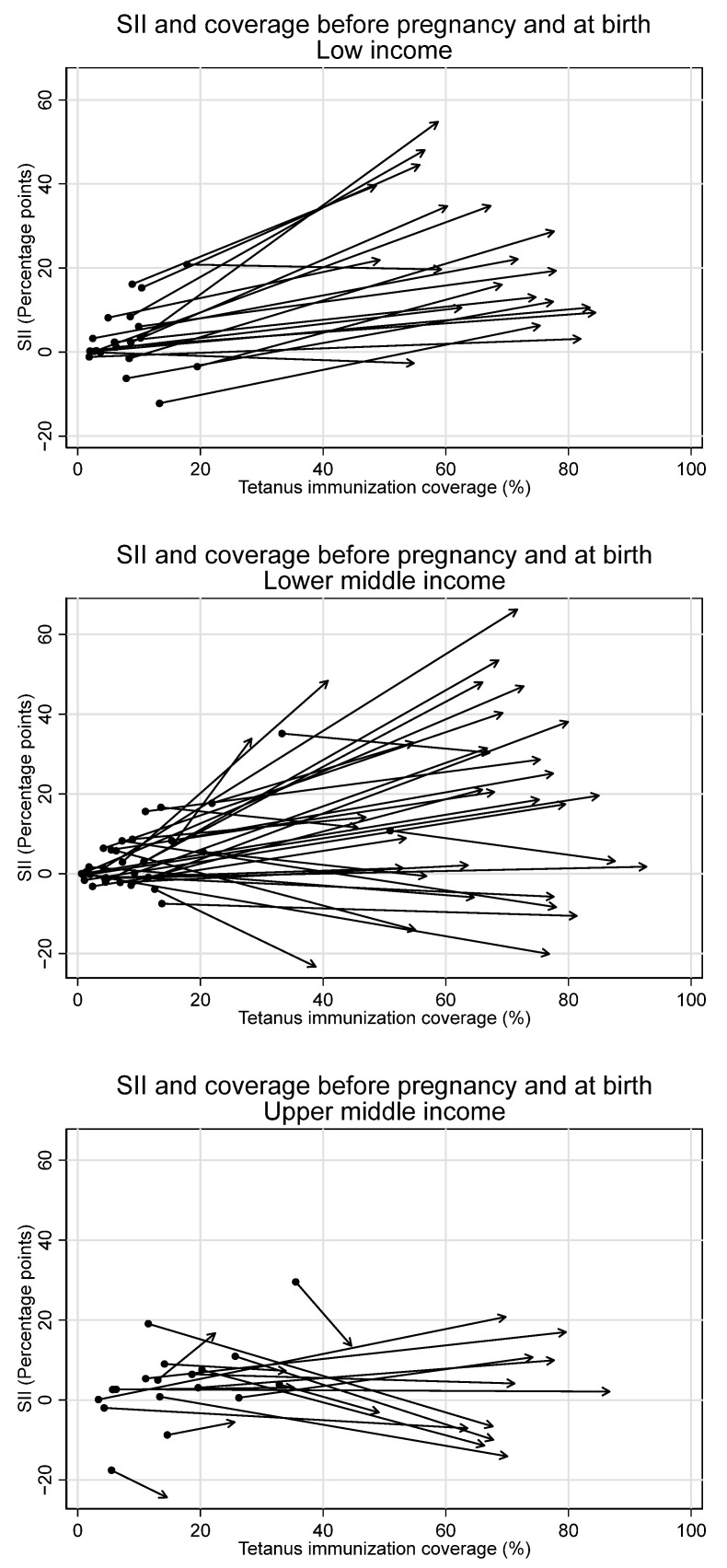
Absolute inequality in maternal tetanus immunization coverage and average coverage level, before pregnancy and at birth, by country income grouping. (Dot represents coverage level and SII in tetanus immunization coverage before pregnancy; arrowhead represents coverage and SII at birth).

**Figure 9 vaccines-12-00431-f009:**
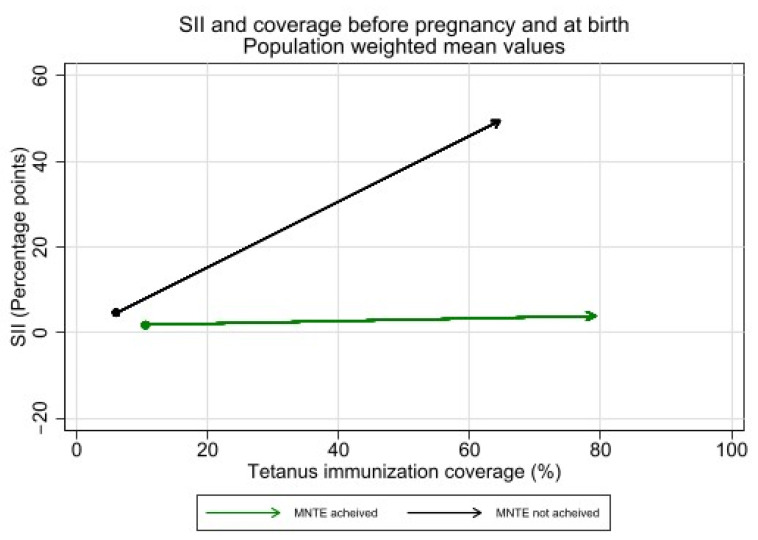
Inequality in maternal tetanus immunization coverage and average coverage level, before pregnancy and at birth, by whether countries had or had not achieved MNTE at time of survey. (Dot represents coverage level and SII in tetanus immunization coverage before pregnancy; arrowhead represents coverage and SII at birth).

**Figure 10 vaccines-12-00431-f010:**
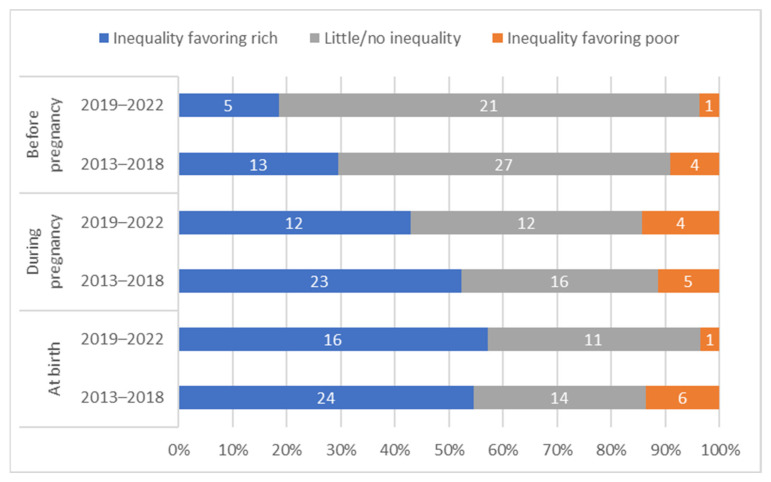
Distribution of countries’ absolute inequality in tetanus immunization coverage before pregnancy, during pregnancy, and at birth; by recency of data collection.

**Table 1 vaccines-12-00431-t001:** Tetanus immunization coverage at birth, before pregnancy, and during pregnancy among first births; most recent DHS or MICS estimates 2013–2022 for 72 included study countries.

Geography	Survey Year	Year of MNTE	Coverage at Birth %	Coverage before Pregnancy %	Coverage during Pregnancy %
*All Countries (median)*			*67.5*	*8.6*	*56.7*
*All Countries (mean *)*			*76.3*	*9.6*	*66.7*
*Low-Income Countries (median)*			*67.3*	*7.9*	*60.6*
*Low-Income Countries (mean *)*			*63.7*	*8.5*	*55.2*
Afghanistan	2015	NA	59.2	17.8	41.5
Benin	2017	2010	67.3	3.0	64.2
Burundi	2016	2009	69.1	19.5	49.6
Central African Republic	2018	NA	58.7	10.2	48.5
Chad	2019	2019	52.8	0.0	52.8
Democratic Republic of the Congo	2017	2019	55.7	10.4	45.3
Ethiopia	2016	2017	56.5	8.6	48.0
Gambia	2019	<2000	54.8	3.7	51.1
Guinea	2018	NA	60.2	8.6	51.6
Guinea-Bissau	2018	2012	77.9	9.9	68.0
Haiti	2016	2017	74.7	6.0	68.7
Liberia	2019	2011	82.0	1.9	80.1
Madagascar	2021	2014	71.7	2.4	69.3
Malawi	2019	2002	83.5	2.7	80.8
Mali	2018	2023	48.6	8.9	39.7
Niger	2021	2016	49.2	5.0	44.3
Rwanda	2019	2004	62.5	2.0	60.6
Sierra Leone	2019	2013	84.4	6.3	78.1
Togo	2017	2005	77.5	7.9	69.6
Uganda	2016	2011	75.4	13.3	62.0
United Republic of Tanzania	2015	2012	77.6	8.4	69.2
*Lower-Middle-Income Countries (median)*			*67.9*	*7.2*	*61.4*
*Lower-Middle-Income Countries (mean *)*			*79.7*	*9.0*	*70.7*
Angola	2015	NA	68.6	10.8	57.8
Bangladesh	2019	2008	87.6	50.9	36.7
Cambodia	2021	2015	63.6	6.9	56.7
Cameroon	2018	2012	69.2	4.2	65.1
Congo	2014	2009	75.3	11.0	64.3
Côte d’Ivoire	2016	2013	67.9	5.5	62.4
Egypt	2014	2007	76.8	0.6	76.2
El Salvador	2014	<2000	81.3	13.7	67.6
Eswatini	2014	<2000	79.5	1.1	78.4
Ghana	2017	2011	66.7	9.2	57.5
Guatemala	2014	<2000	78.0	7.2	70.7
Honduras	2019	<2000	65.9	11.5	54.4
India	2019	2015	92.7	4.4	88.3
Indonesia	2017	2016	56.8	20.4	36.4
Kenya	2022	2018	53.0	4.6	48.4
Kiribati	2018	<2000	53.5	2.4	51.1
Lao People’s Democratic Republic	2017	2013	54.7	21.9	32.8
Lesotho	2018	<2000	77.6	0.9	76.7
Mauritania	2019	2015	46.9	8.9	38.0
Myanmar	2015	2010	75.2	1.7	73.5
Nepal	2022	2005	64.6	1.8	62.8
Nigeria	2021	NA	71.6	4.2	67.4
Pakistan	2017	NA	72.7	1.3	71.3
Papua New Guinea	2016	NA	40.8	7.3	33.5
Philippines	2022	2017	55.1	6.3	48.8
Senegal	2019	2011	84.9	2.0	82.9
Sudan	2014	NA	66.0	4.5	61.4
Timor-Leste	2016	2012	77.5	15.3	62.2
Tunisia	2018	<2000	38.8	12.6	26.2
Viet Nam	2020	2005	79.9	8.7	71.2
Yemen	2013	NA	28.3	15.7	12.6
Zambia	2018	2007	67.2	33.3	33.9
Zimbabwe	2019	2000	45.6	13.6	32.1
*Upper-Middle-Income Countries (median)*			*67.0*	*13.8*	*46.9*
*Upper-Middle-Income Countries (mean *)*			*64.7*	*15.2*	*49.6*
Belize	2015	<2000	63.5	4.3	59.2
Costa Rica	2018	<2000	44.6	35.5	9.1
Cuba	2019	<2000	66.3	20.3	46.0
Dominican Republic	2019	<2000	86.7	6.3	80.4
Fiji	2021	<2000	49.1	32.8	16.2
Gabon	2019	2013	69.7	3.4	66.4
Guyana	2019	<2000	22.4	13.0	9.4
Iraq	2018	2013	67.6	11.5	56.2
Jordan	2017	<2000	25.6	14.6	11.0
Maldives	2016	<2000	74.2	26.3	47.9
Mexico	2015	<2000	71.2	18.6	52.6
Namibia	2013	2001	67.7	25.7	42.1
Paraguay	2016	<2000	79.6	11.1	68.5
Peru	2019	<2000	70.0	13.4	56.6
Samoa	2019	<2000	33.9	14.1	19.8
South Africa	2016	2002	35.2	5.7	29.6
Suriname	2018	<2000	14.5	5.5	9.0
Thailand	2019	<2000	77.6	19.6	58.0

* Means are population weighted and use female population age 15–49 with *aweight* specifications in Stata. MNTE: Maternal and neonatal tetanus elimination; NA: Not achieved. Note that, in some cases, coverage at birth may differ by 0.1 from the sum of coverage before and during pregnancy due to rounding.

**Table 2 vaccines-12-00431-t002:** Absolute wealth-related inequality in tetanus immunization coverage at birth, before pregnancy, and during pregnancy; most recent DHS or MICS estimates 2013–2022 for 72 included study countries.

Geography	Survey Year	SII at Birth			SII before Pregnancy			SII during Pregnancy		
		Estimate	LL	UL	Estimate	LL	UL	Estimate	LL	UL
*All Countries (median)*		*13.3*	*9.0*	*19.5*	*2.4*	*0.3*	*4.2*	*10.6*	*3.4*	*15.6*
*All Countries (mean *)*		*12.8*	*7.7*	*17.9*	*2.4*	*1.0*	*3.7*	*10.5*	*5.3*	*15.6*
*Low-Income Countries (median)*		*19.6*	*11.2*	*33.7*	*1.7*	*−0.1*	*5.8*	*18.4*	*10.6*	*30.4*
*Low-Income Countries (mean *)*		*30.5*	*23.8*	*37.2*	*5.8*	*1.8*	*9.8*	*25.4*	*19.8*	*31.0*
Afghanistan	2015	19.6	−0.5	39.8	20.9	−5.6	47.3	−1.0	−13.7	11.8
Benin	2017	34.8	28.6	41.0	0.3	−2.6	3.3	34.5	30.6	38.4
Burundi	2016	16.0	5.0	27.1	−3.5	−15.9	8.9	19.4	3.5	35.4
Central African Republic	2018	54.8	35.2	74.4	3.4	−2.7	9.5	51.8	30.2	73.5
Chad	2019	32.5	23.9	41.1	--	--	--	32.5	23.9	41.1
Democratic Republic of the Congo	2017	44.6	32.1	57.0	15.3	−0.8	31.4	30.6	6.4	54.9
Ethiopia	2016	48.0	27.7	68.4	8.4	2.2	14.7	40.6	25.3	55.9
Gambia	2019	−2.6	−5.4	0.1	−0.1	−2.9	2.7	−2.6	−6.5	1.4
Guinea	2018	34.7	19.2	50.3	2.3	−2.4	6.9	32.6	19.4	45.7
Guinea-Bissau	2018	19.4	13.4	25.3	6.1	−5.1	17.2	13.3	−2.2	28.8
Haiti	2016	13.1	5.4	20.8	2.4	1.2	3.5	10.7	2.1	19.4
Liberia	2019	3.1	−7.9	14.2	−1.1	−4.3	2.0	4.3	−6.5	15.0
Madagascar	2021	22.1	13.7	30.6	3.3	1.8	4.8	19.0	9.4	28.7
Malawi	2019	10.6	0.9	20.3	0.2	−2.6	2.9	10.4	−1.7	22.5
Mali	2018	39.6	28.0	51.2	16.1	12.5	19.7	25.1	13.3	36.9
Niger	2021	21.9	4.0	39.8	8.2	−0.1	16.4	14.2	4.7	23.8
Rwanda	2019	10.6	5.7	15.4	0.3	−1.6	2.1	10.3	3.9	16.7
Sierra Leone	2019	9.4	5.4	13.4	1.2	−4.0	6.4	8.2	1.2	15.2
Togo	2017	12.0	2.0	22.0	−6.3	−11.9	−0.6	18.2	9.8	26.5
Uganda	2016	6.3	−0.7	13.3	−12.2	−13.5	−10.9	18.4	12.4	24.4
United Republic of Tanzania	2015	28.7	20.5	37.0	−1.5	−8.7	5.7	30.1	18.9	41.2
*Lower-Middle-Income Countries (median)*		*18.6*	*2.5*	*30.0*	*1.4*	*−0.3*	*5.9*	*13.4*	*−1.3*	*19.4*
*Lower-Middle-Income Countries (mean *)*		*11.4*	*3.6*	*19.2*	*1.4*	*−0.2*	*3.0*	*10.0*	*2.1*	*17.8*
Angola	2015	53.5	43.6	63.4	3.0	−3.2	9.3	50.6	42.6	58.6
Bangladesh	2019	3.1	1.7	4.6	10.8	6.5	15.1	−7.7	−10.5	−4.8
Cambodia	2021	2.1	−9.2	13.5	−2.2	−4.5	0.2	4.3	−5.0	13.6
Cameroon	2018	40.3	33.1	47.6	6.4	5.1	7.7	34.5	25.7	43.2
Congo	2014	28.6	9.6	47.6	15.6	8.9	22.4	13.4	−10.5	37.4
Côte d’Ivoire	2016	20.5	−2.9	43.9	6.0	3.9	8.1	14.7	−10.4	39.8
Egypt	2014	−20.1	−29.5	−10.7	0.0	−1.3	1.3	−20.1	−30.7	−9.5
El Salvador	2014	−10.6	−18.1	−3.0	−7.5	−15.3	0.3	−3.1	−10.8	4.6
Eswatini	2014	17.4	−8.2	43.0	−1.6	−5.5	2.3	19.0	−5.2	43.1
Ghana	2017	31.5	20.6	42.4	0.2	−4.0	4.3	31.3	21.5	41.0
Guatemala	2014	−8.4	−29.6	12.9	8.2	7.1	9.4	−16.4	−38.8	6.0
Honduras	2019	21.1	7.7	34.5	−0.9	−3.5	1.7	22.0	9.7	34.2
India	2019	1.8	0.6	2.9	−1.1	−1.7	−0.6	2.9	1.3	4.5
Indonesia	2017	−0.6	−12.3	11.1	5.5	0.0	11.0	−6.1	−13.7	1.4
Kenya	2022	1.4	−3.0	5.9	−1.0	−2.9	0.9	2.4	−1.0	5.8
Kiribati	2018	8.9	−2.9	20.7	−3.2	−9.9	3.6	12.0	3.2	20.8
Lao People’s Democratic Republic	2017	32.9	17.0	48.8	17.7	10.5	24.9	15.8	4.7	26.9
Lesotho	2018	−5.8	−22.8	11.3	−0.4	−2.0	1.2	−5.4	−22.3	11.6
Mauritania	2019	14.1	9.8	18.5	8.6	7.0	10.3	5.7	−0.1	11.4
Myanmar	2015	18.6	8.5	28.7	0.2	−0.8	1.2	18.4	9.1	27.7
Nepal	2022	−5.9	−12.5	0.7	1.7	0.5	2.9	−7.6	−14.2	−1.0
Nigeria	2021	66.3	63.4	69.2	1.4	−1.9	4.8	64.3	60.9	67.7
Pakistan	2017	47.0	35.4	58.6	0.4	−0.6	1.5	46.5	33.6	59.4
Papua New Guinea	2016	48.4	45.1	51.7	2.9	−0.5	6.4	46.0	42.5	49.4
Philippines	2022	−14.1	−21.9	−6.3	5.8	−3.6	15.1	−19.7	−31.6	−7.8
Senegal	2019	19.6	10.2	29.0	−0.2	−1.4	1.1	19.7	9.3	30.0
Sudan	2014	48.0	40.8	55.2	−2.1	−6.4	2.3	49.7	46.6	52.8
Timor-Leste	2016	25.2	16.4	33.9	8.3	−1.3	17.9	17.0	12.4	21.5
Tunisia	2018	−23.3	−36.6	−10.0	−3.9	−8.6	0.9	−19.5	−35.7	−3.3
Viet Nam	2020	38.1	18.3	57.9	−2.9	−10.7	5.0	40.1	25.9	54.2
Yemen	2013	33.9	31.0	36.8	7.7	2.2	13.2	27.8	22.5	33.2
Zambia	2018	30.3	23.2	37.4	35.1	30.0	40.3	−4.9	−13.0	3.1
Zimbabwe	2019	11.6	7.8	15.4	16.7	11.3	22.0	−4.9	−15.0	5.3
*Upper-Middle-Income Countries (median)*		*2.6*	*−6.9*	*10.5*	*3.5*	*0.7*	*7.2*	*−2.0*	*−13.1*	*5.8*
*Upper-Middle-Income Countries (mean *)*		*1.9*	*−1.9*	*5.7*	*5.8*	*2.8*	*8.8*	*−3.8*	*−8.7*	*1.0*
Belize	2015	−7.1	−11.5	−2.6	−2.0	−15.3	11.4	−5.1	−16.2	6.0
Costa Rica	2018	13.6	−8.1	35.3	29.5	1.6	57.5	−17.1	−21.1	−13.2
Cuba	2019	−11.4	−23.8	1.1	7.5	−0.3	15.4	−18.8	−25.2	−12.4
Dominican Republic	2019	2.1	−4.8	9.0	2.7	−2.0	7.4	−0.6	−11.0	9.9
Fiji	2021	−3.1	−25.9	19.7	3.9	−6.5	14.3	−7.0	−25.5	11.5
Gabon	2019	20.8	16.0	25.6	0.1	−0.2	0.5	20.7	15.5	25.8
Guyana	2019	16.7	−1.2	34.7	5.0	−12.1	22.1	11.9	8.3	15.5
Iraq	2018	−6.6	−22.9	9.8	19.1	9.7	28.5	−24.9	−44.0	−5.9
Jordan	2017	−5.5	−8.0	−3.0	−8.7	−15.8	−1.6	3.2	−4.6	11.0
Maldives	2016	10.7	−25.8	47.2	0.6	−4.4	5.5	10.2	−24.5	44.8
Mexico	2015	4.2	−0.4	8.7	6.4	0.0	12.9	−2.3	−12.4	7.8
Namibia	2013	−9.9	−27.3	7.4	10.9	−8.7	30.5	−20.7	−22.8	−18.7
Paraguay	2016	17.0	8.3	25.7	5.4	−0.5	11.3	11.6	1.2	22.0
Peru	2019	−14.1	−29.3	1.1	0.8	−3.6	5.3	−14.9	−34.7	4.8
Samoa	2019	7.3	−17.4	32.0	9.0	−6.0	24.1	−1.7	−16.2	12.8
South Africa	2016	3.1	−2.0	8.3	2.6	−6.4	11.7	0.5	−6.3	7.3
Suriname	2018	−24.4	−38.4	−10.4	−17.5	−26.1	−9.0	−8.7	−19.1	1.7
Thailand	2019	9.9	2.3	17.6	3.1	−0.5	6.7	6.9	−3.5	17.2

* Means are population weighted and use female population age 15–49 with *aweight* specifications in Stata. SII: Slope index of inequality; LL: Lower 95% confidence interval of estimate; UL: Upper 95% confidence interval of estimate.

**Table 3 vaccines-12-00431-t003:** Relative wealth-related inequality in tetanus immunization coverage at birth, before pregnancy, and during pregnancy; most recent DHS or MICS estimates 2013–2022 for 72 included study countries.

Geography	Survey Year	RII at Birth			RII before Pregnancy			RII during Pregnancy		
		Estimate	LL	UL	Estimate	LL	UL	Estimate	LL	UL
*All Countries (median)*		*1.25*	*1.14*	*1.36*	*1.31*	*1.10*	*1.49*	*1.20*	*1.06*	*1.30*
*All Countries (mean *)*		*1.35*	*1.19*	*1.50*	*1.38*	*1.14*	*1.62*	*1.36*	*1.13*	*1.59*
*Low-Income Countries (median)*		*1.37*	*1.18*	*1.78*	*1.26*	*0.99*	*2.60*	*1.35*	*1.18*	*1.83*
*Low-Income Countries (mean *)*		*1.82*	*1.57*	*2.08*	*2.61*	*1.85*	*3.37*	*1.74*	*1.51*	*1.97*
Afghanistan	2015	1.40	0.89	1.91	3.32	0.00	8.35	0.98	0.68	1.28
Benin	2017	1.72	1.52	1.92	1.12	0.02	2.21	1.75	1.62	1.89
Burundi	2016	1.26	1.06	1.47	0.84	0.32	1.35	1.49	1.02	1.96
Central African Republic	2018	2.86	1.57	4.16	1.39	0.62	2.16	3.27	1.50	5.03
Chad	2019	1.89	1.51	2.28	--	--	--	1.89	1.51	2.28
Democratic Republic of the Congo	2017	2.36	1.81	2.92	4.45	0.00	9.17	2.01	1.03	2.99
Ethiopia	2016	2.53	1.29	3.77	2.70	1.09	4.31	2.45	1.39	3.52
Gambia	2019	0.95	0.91	1.00	0.98	0.24	1.71	0.95	0.88	1.02
Guinea	2018	1.83	1.21	2.45	1.30	0.60	2.01	1.93	1.28	2.57
Guinea-Bissau	2018	1.29	1.19	1.39	1.85	0.10	3.60	1.22	0.95	1.49
Haiti	2016	1.19	1.07	1.32	1.49	1.17	1.80	1.17	1.02	1.32
Liberia	2019	1.04	0.90	1.18	0.54	0.00	1.52	1.05	0.91	1.20
Madagascar	2021	1.37	1.20	1.55	3.78	1.45	6.12	1.32	1.13	1.51
Malawi	2019	1.14	1.00	1.27	1.06	0.00	2.17	1.14	0.97	1.31
Mali	2018	2.37	1.59	3.15	6.40	2.26	10.53	1.91	1.22	2.60
Niger	2021	1.57	1.00	2.16	5.27	0.00	15.18	1.38	1.09	1.68
Rwanda	2019	1.18	1.10	1.27	1.14	0.10	2.17	1.19	1.07	1.31
Sierra Leone	2019	1.12	1.06	1.17	1.21	0.24	2.18	1.11	1.01	1.21
Togo	2017	1.17	1.01	1.32	0.45	0.18	0.73	1.30	1.15	1.46
Uganda	2016	1.09	0.98	1.19	0.40	0.35	0.44	1.35	1.21	1.49
United Republic of Tanzania	2015	1.47	1.30	1.64	0.83	0.13	1.54	1.57	1.30	1.84
*Lower-Middle-Income Countries (median)*		*1.29*	*1.03*	*1.56*	*1.31*	*0.92*	*1.70*	*1.27*	*0.98*	*1.44*
*Lower-Middle-Income Countries (mean *)*		*1.31*	*1.08*	*1.55*	*1.14*	*0.92*	*1.37*	*1.35*	*0.98*	*1.72*
Angola	2015	2.43	1.81	3.05	1.32	0.56	2.09	2.63	2.01	3.24
Bangladesh	2019	1.04	1.02	1.05	1.24	1.13	1.35	0.81	0.75	0.87
Cambodia	2021	1.03	0.85	1.22	0.73	0.49	1.00	1.08	0.90	1.26
Cameroon	2018	1.87	1.60	2.13	4.65	2.58	6.71	1.74	1.46	2.03
Congo	2014	1.48	1.06	1.91	4.23	0.51	7.95	1.23	0.76	1.71
Côte d’Ivoire	2016	1.36	0.88	1.84	3.00	1.37	4.63	1.27	0.76	1.78
Egypt	2014	0.77	0.66	0.87	1.05	0.00	3.22	0.76	0.65	0.88
El Salvador	2014	0.88	0.80	0.96	0.58	0.19	0.96	0.96	0.85	1.06
Eswatini	2014	1.25	0.82	1.68	0.23	0.00	1.14	1.28	0.86	1.70
Ghana	2017	1.63	1.33	1.94	1.02	0.56	1.48	1.76	1.41	2.10
Guatemala	2014	0.90	0.65	1.15	3.15	2.54	3.76	0.79	0.53	1.06
Honduras	2019	1.38	1.11	1.66	0.92	0.71	1.14	1.51	1.17	1.85
India	2019	1.02	1.01	1.03	0.77	0.67	0.87	1.03	1.01	1.05
Indonesia	2017	0.99	0.79	1.19	1.31	0.96	1.66	0.84	0.66	1.03
Kenya	2022	1.03	0.94	1.11	0.81	0.49	1.12	1.05	0.98	1.12
Kiribati	2018	1.18	0.92	1.45	0.27	0.00	0.90	1.27	1.06	1.47
Lao People’s Democratic Republic	2017	1.87	1.19	2.54	2.27	1.35	3.19	1.62	1.01	2.24
Lesotho	2018	0.93	0.72	1.14	0.66	0.00	1.68	0.93	0.72	1.14
Mauritania	2019	1.35	1.22	1.49	2.65	1.99	3.32	1.16	0.98	1.34
Myanmar	2015	1.29	1.11	1.46	1.14	0.49	1.80	1.29	1.12	1.46
Nepal	2022	0.91	0.82	1.01	2.55	0.67	4.44	0.89	0.79	0.98
Nigeria	2021	3.21	2.92	3.50	1.40	0.40	2.39	3.20	2.88	3.51
Pakistan	2017	2.05	1.56	2.55	1.39	0.26	2.52	2.05	1.50	2.61
Papua New Guinea	2016	3.66	3.25	4.07	1.50	0.73	2.27	4.47	3.96	4.97
Philippines	2022	0.77	0.67	0.88	2.52	0.00	5.46	0.66	0.49	0.83
Senegal	2019	1.27	1.11	1.42	0.93	0.37	1.48	1.27	1.10	1.45
Sudan	2014	2.23	1.92	2.54	0.63	0.02	1.25	2.44	2.29	2.60
Timor-Leste	2016	1.40	1.23	1.57	1.72	0.65	2.80	1.32	1.21	1.42
Tunisia	2018	0.54	0.36	0.73	0.73	0.47	1.00	0.47	0.19	0.75
Viet Nam	2020	1.68	1.14	2.21	0.72	0.06	1.37	1.83	1.36	2.30
Yemen	2013	3.51	3.04	3.98	1.64	1.10	2.18	10.23	5.03	15.42
Zambia	2018	1.60	1.41	1.78	3.03	2.47	3.59	0.86	0.66	1.07
Zimbabwe	2019	1.29	1.18	1.40	3.49	1.83	5.15	0.86	0.59	1.13
*Upper-Middle-Income Countries (median)*		*1.04*	*0.87*	*1.22*	*1.43*	*1.04*	*1.57*	*0.94*	*0.65*	*1.17*
*Upper-Middle-Income Countries (mean *)*		*1.04*	*0.97*	*1.10*	*1.78*	*1.16*	*2.41*	*0.95*	*0.84*	*1.06*
Belize	2015	0.89	0.83	0.96	0.63	0.00	2.44	0.92	0.75	1.09
Costa Rica	2018	1.36	0.65	2.07	2.36	0.06	4.66	0.14	0.05	0.24
Cuba	2019	0.84	0.68	1.00	1.45	0.89	2.01	0.66	0.57	0.75
Dominican Republic	2019	1.02	0.94	1.11	1.54	0.30	2.77	0.99	0.86	1.12
Fiji	2021	0.94	0.51	1.37	1.13	0.78	1.48	0.65	0.00	1.34
Gabon	2019	1.35	1.26	1.45	1.04	0.92	1.16	1.37	1.26	1.48
Guyana	2019	2.13	0.20	4.05	1.47	0.00	3.51	3.63	2.03	5.23
Iraq	2018	0.91	0.69	1.12	5.50	1.91	9.09	0.64	0.42	0.85
Jordan	2017	0.81	0.73	0.88	0.55	0.30	0.80	1.34	0.32	2.36
Maldives	2016	1.16	0.59	1.72	1.02	0.83	1.21	1.24	0.36	2.12
Mexico	2015	1.06	0.99	1.13	1.41	0.93	1.90	0.96	0.77	1.14
Namibia	2013	0.86	0.63	1.09	1.53	0.43	2.64	0.61	0.57	0.64
Paraguay	2016	1.24	1.10	1.38	1.63	0.70	2.56	1.19	1.00	1.37
Peru	2019	0.82	0.64	1.00	1.07	0.71	1.42	0.77	0.49	1.04
Samoa	2019	1.24	0.33	2.15	1.90	0.00	3.83	0.92	0.26	1.58
South Africa	2016	1.09	0.94	1.25	1.59	0.00	4.16	1.02	0.78	1.25
Suriname	2018	0.18	0.00	0.39	0.04	0.00	0.09	0.38	0.00	0.91
Thailand	2019	1.14	1.02	1.25	1.17	0.95	1.39	1.13	0.92	1.33

* Means are population weighted and use female population age 15–49 with *aweight* specifications in Stata. RII: Relative index of inequality; LL: Lower 95% confidence interval of estimate; UL: Upper 95% confidence interval of estimate.

## Data Availability

All analyses were carried out using publicly available datasets that can be obtained directly from the DHS (dhsprogram.com (accessed on 29 September 2023)) and the MICS (mics.unicef.org (accessed on 29 September 2023)) websites. Datasets are continuously sourced and updated by the International Center for Equity in Health (equidade.org) as they are released. Analyses used the latest available dataset versions as of 29 September 2023.

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
