# Peer review of "Comparison of Wealth-Related Inequality in Tetanus Vaccination Coverage before and during Pregnancy: A Cross-Sectional Analysis of 72 Low- and Middle-Income Countries"

_vaccines, 2024, doi:10.3390/vaccines12040431_

Round 1

Reviewer 1 Report

Comments and Suggestions for Authors

The research reported in this paper is well done and the findings contribute to knowledge. Here are a couple of item to attend to in a revision.

First, to give readers information up front about the focus of the contents and research in the paper, it would be good to emphasize "low-income to upper-middle income" countries. For example, in the title of the paper "a cross-sectional analysis of 72 low-income to upper-middle income countries". And then throughout the paper in various places in the text.

Second, also to improve quick understanding on the part of readers, it would be good to note that the entries in some of the tables and figures as well as in some places in the text are percentages. For example, in the title of Table 1, you could add percentages as in "Tetanus immunization coverage percentages at birth". And in Figures 7, 8, and 9, you could indicate that the SII vertical axis and Coverage horizontal axis measures are percentages. 

These edits are minor, but they will improve comprehension of the text of the paper and its findings by readers.

Author Response

We thank the reviewer for their time and expertise. Please see the attachment for response.

REVIEWER 1

The research reported in this paper is well done and the findings contribute to knowledge. Here are a couple of items to attend to in a revision.

We appreciate the reviewer’s time and expertise in reviewing the manuscript.

First, to give readers information up front about the focus of the contents and research in the paper, it would be good to emphasize "low-income to upper-middle income" countries. For example, in the title of the paper "a cross-sectional analysis of 72 low-income to upper-middle income countries". And then throughout the paper in various places in the text.

Thank you for flagging this helpful clarification. We have revised the title, abstract (lines 28-29), and text throughout (lines 110, 232, 429) to clarify that this analysis is limited to low- and middle-income countries. We have also added to the limitations (lines 603-616) that this analysis has been limited to low- and middle-income countries where nationally-representative surveys have been recently conducted, reducing generalizability.

Second, also to improve quick understanding on the part of readers, it would be good to note that the entries in some of the tables and figures as well as in some places in the text are percentages. For example, in the title of Table 1, you could add percentages as in "Tetanus immunization coverage percentages at birth". And in Figures 7, 8, and 9, you could indicate that the SII vertical axis and Coverage horizontal axis measures are percentages. 

This is duly noted; we have added “%” to relevant column headings in Table 1, and updated axis titles in figures 7, 8, and 9 to “Tetanus immunization coverage (%)” and “SII (Percentage points)” accordingly.

These edits are minor, but they will improve comprehension of the text of the paper and its findings by readers.

Reviewer 2 Report

Comments and Suggestions for Authors

This study determined the inequality in TTCV vaccination during pregnancy in 72 countries, especially in low- and lower-middle income countries.

1, The abstract might be too long. It should be revised to highlight possible points and findings in the study.

2, As stated in the study limitations, this study was an ecological study, utilizing public data that were collected at inconsistent points of time (even a decade ago) to determine potential association between the TTCV uptake and inequality. The findings are interesting; however, the analysis might not be robust. I wonder if the authors may conduct possible sub-group analysis, by using the data reliably collected in some countries.

3, This study determined the change in the inequality and explored its influence on the TTCV vaccination. In the analysis, the inequality was only defined as wealth-related. However, the inequality is complex. For example, some low-income countries had significantly lower TTCV coverage, which might have more influence than wealth. I mean most wealthy families in those countries also had limited access to or awareness of TTCV vaccination. Please consider how to improve the analysis or conduct further analysis in the manuscript.

Author Response

We thank the reviewer for their time and expertise. Please see the attachment for response.

REVIEWER 2

This study determined the inequality in TTCV vaccination during pregnancy in 72 countries, especially in low- and lower-middle income countries.

We appreciate the reviewer’s time and expertise in reviewing the manuscript.

1, The abstract might be too long. It should be revised to highlight possible points and findings in the study.

Thank you for noting this, we have reduced the abstract to 250 words, with the abstract now more focused on the analyses and findings of the study.

2, As stated in the study limitations, this study was an ecological study, utilizing public data that were collected at inconsistent points of time (even a decade ago) to determine potential association between the TTCV uptake and inequality. The findings are interesting; however, the analysis might not be robust. I wonder if the authors may conduct possible sub-group analysis, by using the data reliably collected in some countries.

We appreciate the reviewer’s concern regarding the robustness of findings from data which are now up to 10 years old. Unfortunately, representative longitudinal data from these settings are not readily available, and these large-scale representative surveys have (necessary) lag times for public availability. To examine whether the findings are sensitive to the time period of data used, however, we have added a post-hoc sensitivity analysis to the manuscript which replicates several key analyses with the subset of countries with data <=5 years old (2019-2022, n=28), and contrasts findings with those from data >5 years old (2013-2018). We find that, as in the full sample, the majority of PAB coverage occurs during pregnancy in most countries with recent data (25/28, 89%). We find significantly lower average absolute wealth-related inequality in the more recent data before pregnancy (average SII for recent data 0.8 vs 4.9 for older data, p=0.003) and significantly lower average inequality at birth (average SII 8.7 vs 19.3, p=0.04), though no significant difference in average inequality during pregnancy (p=0.20). We find consistent patterns of inequality whereby a similar proportion of countries have statistically significant inequalities before pregnancy, during pregnancy, and at birth (see below figure).

In both older and more recent data, we observe significant average inequality favoring the wealthiest during the pregnancy time period (average SII for recent data 7.9, 95% CI 0.3-15.5; average SII for older data 14.6, 95% CI 7.4-21.8). Note that these findings should not imply temporal trends, as each time period contains a unique group of countries. However, consistent findings of significant inequality during pregnancy and consistent evidence of greater inequality during pregnancy compared to before pregnancy suggest that the overall conclusions of the study are not sensitive to a 5-year rather than 10-year analysis time frame.

We also note that the association between TTCV coverage and inequality in TTCV coverage is presented as an association, with no suggestion of causality; as we are evaluating coverage and inequality in coverage at the same point in time, no assessment or assumption of temporality is possible. The primary analyses, presenting inequalities in coverage level, in fact involve no tests of association at all, reducing issues of potential ecological fallacy.

We have added this sensitivity analysis to the methods (lines 211-214), results (lines 404-427), and limitations (lines 601-606). We have also added text to the limitations section regarding the limited generalizability of this survey given data that may now be up to 10 years out of date (lines 603-616). We have also added more explicit discussion of the limits of cross-sectional ecological analyses to the limitations, as well as the need for further work to understand the factors which lead to inequalities in coverage (lines 617-625).

3, This study determined the change in the inequality and explored its influence on the TTCV vaccination. In the analysis, the inequality was only defined as wealth-related. However, the inequality is complex. For example, some low-income countries had significantly lower TTCV coverage, which might have more influence than wealth. I mean most wealthy families in those countries also had limited access to or awareness of TTCV vaccination. Please consider how to improve the analysis or conduct further analysis in the manuscript.

This point is well taken, and has multiple components. Firstly, indeed, wealth is only one of many dimensions of inequality which may influence TTCV uptake at the individual level. Though beyond the scope of the current analyses, previous work has demonstrated significant inequalities in TTCV coverage by other factors including maternal age, maternal education, and urban/rural residence. As wealth and these factors are often related (e.g. greater wealth and greater maternal education are associated), inequalities observed in one domain may in fact be driven by differences in the other. We cannot tell from these analyses what the relative contribution of various dimensions of inequality are, but we have added note to the limitations that these analyses reflect only one dimension of inequality, and that future work which jointly considers multiple dimensions would allow for greater understanding of the full complexities of inequalities in immunization coverage in a population (lines 617-625).

Regarding the issue of overall TTCV coverage levels and wealth-related TTCV inequality, we can test this via a correlation between national coverage levels and SII. In doing so, we do not observe a significant correlation between coverage levels and inequality during pregnancy (r=0.21, p=0.07) nor at birth (r=0.05, p=0.71). However, we do observe a significant association between TTCV coverage levels and inequalities before birth, suggesting that higher coverage before pregnancy is correlated with higher inequality in TTCV before pregnancy (r=0.49, p<0.001). As we find that the majority of inequality in TTCV is introduced during pregnancy, where we observe no association with TTCV coverage, this suggests that coverage levels alone are not driving inequalities. In other words, we find consistent evidence of individual wealth-related inequality in TTCV coverage, particularly during pregnancy, across a range of settings which have differing overall TTCV coverage levels. The correlation between coverage levels and inequality during pregnancy is now included in the results text (lines 360-362).

Reviewer 3 Report

Comments and Suggestions for Authors

I was invited to revise the paper entitled "Comparison of wealth-related inequality in tetanus vaccination coverage before and during pregnancy: a cross-sectional analysis of 72 countries". It was a mulcenter cross-sectional study aimed to evaluate the equality in vaccination coverages towards tetanus among pregnant women i low and middle income countries.

Despite the topic is relevant and the paper was well written, I have some observations:

- Introduction is well written and clearly presented;

- About methods, it is unclear the data sources. In particular in lines 109-110 Authors reported information of two different studies (ref 14 and 15). As actually presented, it is totally unclear. Authors should deeply describe data sources;

- Ref 15 was too old. How can results from a study dated 2021 be considered for the actual context? Authors should better clarify data sources;

- Authors should report more detailed information on indicators stated in line 121;

- Did Authors considered the mean age at pregnancy reported by each country? older women could frequently leave the DT booster or they could report a lower IG titer compared to younger;

- Among discussions, Authors should also consider the impact of different vaccination schedule between countries;

- Among limitations, I suggest to add the ecologial study design that can't take into account several confounders.

Author Response

We thank the reviewer for their time and expertise. Please see the attachment for response.

REVIEWER 3

I was invited to revise the paper entitled "Comparison of wealth-related inequality in tetanus vaccination coverage before and during pregnancy: a cross-sectional analysis of 72 countries". It was a multicenter cross-sectional study aimed to evaluate the equality in vaccination coverages towards tetanus among pregnant women in low- and middle-income countries.

We appreciate the reviewer’s time and expertise in reviewing the manuscript.

Despite the topic is relevant and the paper was well written, I have some observations:

- Introduction is well written and clearly presented;

Thank you!

- About methods, it is unclear the data sources. In particular in lines 109-110 Authors reported information of two different studies (ref 14 and 15). As actually presented, it is totally unclear. Authors should deeply describe data sources;

Thank you for noting this previous lack of clarity. The data in fact come from 72 sources, all of which are either Demographic and Health Surveys (DHS) and Multiple Indicator Cluster Surveys (MICS). References 14/15 (with revision, now references 11/12) detail the general survey methodologies for DHS and MICS approaches; citation for all 72 data sources would unfortunately exceed the citation limits for the journal. Clarification on the nature of DHS and MICS can be found in the introduction lines 100-106; revised clarification on the data sources can be found in the methods section lines 109-120 and in the results section lines 232-237. The specific country-years of included surveys can be found in Table 1 (starting after line 263).

- Ref 15 was too old. How can results from a study dated 2012 be considered for the actual context? Authors should better clarify data sources;

While we agree that the reality in immunization from 2012 to 2024 has changed dramatically, we note that reference 15 (now reference 12 in revised draft) details the methodology for conducting Demographic and Health Surveys generally; this general methodology has in fact not changed since 2012. The data sources themselves, as noted in the response to the point immediately above, are from the specific country-year of data from DHS or MICS as outlined in Table 1. Individually citing all 72 surveys/data sources is unfortunately not feasible, but we confirm that all 72 included surveys involved data collection in 2013 or later. We note that key studies for providing broader context in the introduction (current references 2, 3, 10) were published within the past year.

- Authors should report more detailed information on indicators stated in line 121;

We have added clarification in revised lines 122-123 that all outcome measures are explained in full in the following section. Indicators (e.g. outcome measures, we have modified this text for clarity) used in analysis are detailed in lines 125-159, section ‘2.2 Outcome measures’.

- Did Authors considered the mean age at pregnancy reported by each country? older women could frequently leave the DT booster or they could report a lower IG titer compared to younger;

This point is well taken. While inclusion of maternal age in statistical analyses is outside of the scope of the current analyses, we now note in the discussion that age of pregnancy may influence the time of receipt of vaccination in multiple ways. In particular, older women have had greater time to receive any needed adult booster doses, increasing likelihood of pre-pregnancy coverage, though may have greater recall bias for doses received at younger ages, underestimating earlier dose receipt (note that IG titer data is not collected as part of the included DHS or MICS studies). Our analyses limited to primiparity will somewhat reduce the issue of differential age at birth, though we acknowledge that there is still significant variation in average age at first birth across the examined settings. As the focus of analyses involves within-country inequalities, our primary findings should remain robust to these differences in average age across countries, though these differences affect conclusions drawn for any specific context. We also note maternal age as a potentially confounding factor, and one for future research to consider, in lines 618-625. We have further added consideration of this point to the limitations lines 572-578 as follows:

Additionally, average age at first birth also varies substantially across countries, affecting how long women have the opportunity to receive needed adult boosters prior to pregnancy and the length of time since early childhood and adolescent doses most subject to recall bias. Any country-specific application of findings should take into consideration the broader fertility and immunization system context in that country, inclusive of childhood immunization coverage and evolving tetanus vaccination strategies and schedules, to generate the most appropriate conclusions for that setting.

- Among discussions, Authors should also consider the impact of different vaccination schedule between countries;

This point is duly noted. We now discuss the importance of and variation in the TTCV schedule in discussion lines 436-448, highlighting that there is variation in schedules and that the change in schedules over time toward greater adherence of a 6-dose childhood and adolescent schedule will necessarily change the patterns in PAB coverage and inequality, specifically towards greater coverage in the pre-pregnancy time period. We have also added additional text to the limitations noting the importance of considering country-specific context in vaccination schedule (and other related vaccine system factors) when drawing conclusions for any specific individual country in lines 569-578.

- Among limitations, I suggest to add the ecological study design that can't take into account several confounders.

We fully agree that this is a limitation of the study and that simultaneous consideration of other factors, including potential confounders, is an area for future research. We have added new paragraph in the limitations, lines 617-625, to address this point as follows:

Finally, the analyses presented here are cross-sectional and limited to the examination of a single dimension of inequality, household wealth. Other dimensions such as maternal age, maternal education, or urban/rural residence, may also be meaningful determinants of tetanus immunization coverage, and when considered jointly, may in fact be greater drivers of inequality than wealth alone. It is beyond the scope of these analyses to examine additional determinants of coverage, dimensions of inequality, or the relative contribution of multiple dimensions of inequality; future work in these areas would allow for greater understanding of the full complexities of inequalities in immunization coverage.

Reviewer 4 Report

Comments and Suggestions for Authors

Estimated Authors,

I've read with great interest the your article on the comparison of wealth-related inequality in tetanus vaccination coverage before and during pregnancy. In this cross-sectional analysis of 72 countries, Authors were able to identify an interesting trend, with nearly half of countries showing improved trends, while the other half showed a worsening of the health inequalities associated with tetanus vaccination status.

In my opinion, the present paper could be accepted for publication after some improvements, the most significant being providing a link between estimates on vaccination status (as provided by the current status of the paper) and available estimates on maternal and neonatal tetanus. 

Authors should also disclose that their estimates may have been affected, in terms of consistence with real-world experience, by the COVID-19 pandemic and a precautionary approach is therefore required.

After such improvements the paper could be accepted for publication.

Comments on the Quality of English Language

The paper appears of sufficient quality for an eventual publication.

Author Response

We thank the reviewer for their time and expertise. Please see the attachment for response.

REVIEWER 4

Estimated Authors,

I've read with great interest the your article on the comparison of wealth-related inequality in tetanus vaccination coverage before and during pregnancy. In this cross-sectional analysis of 72 countries, Authors were able to identify an interesting trend, with nearly half of countries showing improved trends, while the other half showed a worsening of the health inequalities associated with tetanus vaccination status.

We appreciate the reviewer’s time and expertise in reviewing the manuscript.

In my opinion, the present paper could be accepted for publication after some improvements, the most significant being providing a link between estimates on vaccination status (as provided by the current status of the paper) and available estimates on maternal and neonatal tetanus. 

Thank you for noting this. We consider this linkage between vaccination status and MNT prevalence in part via our analyses examining countries which have and have not achieved MNTE (see results lines 377-403); by definition, the prevalence of maternal and neonatal tetanus will be so minimal as to negate the possibility of statistical comparison between prevalence and vaccination coverage within countries which have achieved elimination. This leaves 14 countries which had not achieved MNTE by time of survey, within which we observe significantly greater inequality in immunization in pregnancy compared to those which had achieved MNTE. To address your point more directly, we have now added a statement of comparison of vaccination coverage among countries which had and had not achieved MNTE, noting that countries which had not achieved MNTE had significantly lower TTCV coverage at each time point (lines 386-389):

Average TTCV coverage was significantly lower among countries which had not achieved MNTE relative to countries which had achieved MNTE before pregnancy (6.0% vs 10.5%, p=0.03), during pregnancy (58.6% vs 68.7%, p=0.04), and at birth (64.6% vs 79.1%, p<0.001).

Relatedly, based on feedback from other reviewers, we have added greater clarification that our estimates for coverage are for primiparous women only, and therefore should not be considered reflective of coverage for multiparous women or for the population as a whole. Our response to this point of the non-representativeness of our coverage estimates is as follows:

It is reasonable to assume that each pregnancy is an opportunity for immunization, and thus we would expect increasing vaccination protection at birth with increasing birth order (e.g. the mother is more likely to have interacted with the health system and received any necessary/missing TTCV vaccinations during previous pregnancies and births). Conversely, we would expect to see somewhat lower vaccination protection at birth in this sample relative to the birth sample as a whole. We compared our PAB coverage estimates to those from similar previous work that was not limited to primiparous women (https://www.ncbi.nlm.nih.gov/pmc/articles/PMC10146835/), were we see evidence of slightly higher PAB for the all-parity sample: median coverage was 69.1% in the all-parity study, whereas it was 67.5% in this study. However, the included country-years of data in each analysis are not identical, which may drive some of this difference. The restriction of the sample to primiparity has now been highlighted in the limitations (lines 558-563), with this comparison and likely underestimation of true PAB noted. We have also updated the title of Table 1 (lines 262-263), where PAB coverage estimates are provided, to clarify that this is among first births only. As the focus of the analysis was primarily on inequalities in coverage, rather than coverage levels, we feel this limitation does not significantly undercut our results.

Authors should also disclose that their estimates may have been affected, in terms of consistence with real-world experience, by the COVID-19 pandemic and a precautionary approach is therefore required.

This point is well taken, we have added explicit discussion of the potential impacts of the COVID-19 pandemic to a new paragraph discussing the limited generalizability of findings, specifically lines 611-616. The added limitations paragraph in full reads:

Fourth, this research is subject to the limits of the available data, including only a sample of low- and middle-income countries with nationally-representative surveys and data which may be up to 10 years old. The coverage and timing of TTCV dose receipt are also subject to immunization card ownership or recall. Patterns of coverage and inequality may have changed since the time of survey, and survey estimates may underrepresent true coverage. The sensitivity analyses conducted to examine data collected within five years and greater than five years ago suggest that the conclusions of these analyses are not sensitive to a five-year rather than ten-year analysis time frame, however, findings may still be changing over time. In particular, the COVID-19 pandemic has had large impacts on immunization systems and healthcare service delivery generally; children, adolescents, and women who missed TTCV doses due to the pandemic and resultant health system disruptions should be targeted for catch-up doses to sustain coverage and minimize inequality. Studies which replicate these analyses using the most up-to-date data in a given context will be most valuable for elucidating the current state of inequality.

After such improvements the paper could be accepted for publication

Reviewer 5 Report

Comments and Suggestions for Authors

I have read this paper with great interest, and value the analysis as conducted. I do have however some specific reflections

First, there are no ‘first world’ countries in the analysis. This is likely fair, but should be somewhat better reflected in the title, and abstract (world bank, income grouping low income, lower middle, upper middle, no high income countries).  

Related to the ‘concept’ of the paper with focus on vaccination, it is my personal assessment that there is association with ‘other types of appropriate care, like umbilical cord care’. Can the authors provide their opinion (association versus causality). You somewhat refer to this (eg line 58, ‘access to adequate health services’). Tetanos ‘protection’ goes beyond immunization alone.

Related to the focus on primigravida: although I understand the rationale, I assume that this somewhat ‘overestimated’ the immunization ‘failure’, if/once pregnancy becomes an event to ‘induce’ vaccination practices.

Specific

Abstract: DHS or MICS, should like be in full at first mentioning.

Figure 1: Not sure if both ‘grey’ colors sufficiently well discriminate ?

Author Response

We thank the reviewer for their time and expertise. Please see the attachment for response.

REVIEWER 5

I have read this paper with great interest, and value the analysis as conducted. I do have however some specific reflections

We appreciate the reviewer’s time and expertise in reviewing the manuscript.

First, there are no ‘first world’ countries in the analysis. This is likely fair, but should be somewhat better reflected in the title, and abstract (world bank, income grouping low income, lower middle, upper middle, no high income countries).  

Thank you for flagging this helpful clarification. We have revised the title, abstract (lines 28-29), and text throughout (lines 110, 232, 429) to clarify that this analysis is limited to low- and middle-income countries. We have also added to the limitations (lines 603-616) that this analysis has been limited to low- and middle-income countries where nationally-representative surveys have been recently conducted, reducing generalizability.

Related to the ‘concept’ of the paper with focus on vaccination, it is my personal assessment that there is association with ‘other types of appropriate care, like umbilical cord care’. Can the authors provide their opinion (association versus causality). You somewhat refer to this (e.g. line 58, ‘access to adequate health services’). Tetanus ‘protection’ goes beyond immunization alone.

This point is well taken. The protection afforded by safe and sterile delivery and umbilical cord care is now explicitly mentioned in the abstract (lines 20-21) and introduction (lines 69-70). We have expanded our discussion to address the overlap between immunization and healthcare utilization, specifically facility delivery and skilled birth attendance, both of which can reduce risk of tetanus exposure through appropriate, safe, and sterile birth practices (lines 468-474). As the outcome measure of this analysis was immunization coverage, rather than total protection from tetanus or tetanus incidence, this association between healthcare utilization, tetanus immunization, and safe birth practices should not change the study findings, but it does have important implications for the interpretation of results.

Related to the focus on primigravida: although I understand the rationale, I assume that this somewhat ‘overestimated’ the immunization ‘failure’, if/once pregnancy becomes an event to ‘induce’ vaccination practices.

Yes, it is reasonable to assume that each pregnancy is an opportunity for immunization, and thus we would expect increasing vaccination protection at birth with increasing birth order (e.g. the mother is more likely to have interacted with the health system and received any necessary/missing TTCV vaccinations during previous pregnancies and births). Conversely, we would expect to see somewhat lower vaccination protection at birth in this sample relative to the birth sample as a whole. We compared our PAB coverage estimates to those from similar previous work that was not limited to primiparous women (https://www.ncbi.nlm.nih.gov/pmc/articles/PMC10146835/), were we see evidence of slightly higher PAB for the all-parity sample: median coverage was 69.1% in the all-parity study, whereas it was 67.5% in this study. However, the included country-years of data in each analysis are not identical, which may drive some of this difference. The restriction of the sample to primiparity has now been highlighted in the limitations (lines 558-563), with this comparison and likely underestimation of true PAB noted. We have also updated the title of Table 1 (lines 262-263), where PAB coverage estimates are provided, to clarify that this is among first births only. As the focus of the analysis was primarily on inequalities in coverage, rather than coverage levels, we feel this limitation does not significantly undercut our results.

Specific

Abstract: DHS or MICS, should like be in full at first mentioning.

We have now spelled out Demographic and Health Surveys and Multiple Indicator Cluster Surveys in the abstract (lines 29-30), as well as in the introduction (lines 101-102), where first used in text.

Figure 1: Not sure if both ‘grey’ colors sufficiently well discriminate?

Thank you for noting this, however, this shading is standard for maps produced by the World Health Organization and will be maintained. Note that only two ‘Not applicable’ (darker grey) areas are visible at this resolution.

Round 2

Reviewer 2 Report

Comments and Suggestions for Authors

The manuscript has been well revised.